# Dissection of the *in vitro* developmental program of *Hammondia hammondi* reveals a link between stress sensitivity and life cycle flexibility in *Toxoplasma gondii*

Sarah L Sokol[1†], Abby S Primack[1†], Sethu C Nair[1], Zhee S Wong[1], Maiwase Tembo[1], Shiv K Verma[2], Camila K Cerqueira-Cezar[2], JP Dubey[2], Jon P Boyle[1*]

[1]Department of Biological Sciences, Dietrich School of Arts and Sciences, University of Pittsburgh, Pittsburgh, United States; [2]Animal Parasitic Diseases Laboratory, Beltsville Agricultural Research Center, Agricultural Research Service, U.S. Department of Agriculture, Beltsville, United States

*For correspondence:
boylej@pitt.edu

†These authors contributed equally to this work

Competing interests: The authors declare that no competing interests exist.

**Abstract** Most eukaryotic parasites are obligately heteroxenous, requiring sequential infection of different host species in order to survive. *Toxoplasma gondii* is a rare exception to this rule, having a uniquely facultative heteroxenous life cycle. To understand the origins of this phenomenon, we compared development and stress responses in *T. gondii* to those of its its obligately heteroxenous relative, *Hammondia hammondi* and have identified multiple *H. hammondi* growth states that are distinct from those in *T. gondii*. Of these, the most dramatic difference was that *H. hammondi* was refractory to stressors that robustly induce cyst formation in *T. gondii*, and this was reflected most dramatically in its unchanging transcriptome after stress exposure. We also found that *H. hammondi* could be propagated *in vitro* for up to 8 days post-excystation, and we exploited this to generate the first ever transgenic *H. hammondi* line. Overall our data show that *H. hammondi* zoites grow as stringently regulated, unique life stages that are distinct from *T. gondii* tachyzoites, and implicate stress sensitivity as a potential developmental innovation that increased the flexibility of the *T. gondii* life cycle.
DOI: https://doi.org/10.7554/eLife.36491.001

## Introduction

*Toxoplasma gondii*, the causative agent of toxoplasmosis, is a globally ubiquitous Apicomplexan parasite capable of infecting all warm-blooded animals, including approximately one-third of the human population (*Dubey and Beattie, 1988*; *Dubey and Sreekumar, 2003*; *Pappas et al., 2009*; *Jones et al., 2001*). Although generally asymptomatic in immunocompetent individuals, *T. gondii* is capable of causing disease in the immunocompromised (including HIV/AIDS patients), developing neonates, and occasionally healthy individuals (*Luft et al., 1993*; *Remington and Klein, 2001*; *Carme et al., 2002*; *Grigg et al., 2001*). The global ubiquity of *T. gondii* is due, at least in part, to its broad host range and ability to persist undetected in an immunocompetent host where it is either capable of being transmitted to a new intermediate host or recrudescing in the original host (*Marty et al., 2002*; *Takashima et al., 2008*). These characteristics are lacking in the nearest extant relative of *T. gondii*, *Hammondia hammondi*. Despite sharing the vast majority of their genes in near perfect synteny (*Walzer et al., 2013*) and the same definitive felid host (*Dubey and Sreekumar,*

**eLife digest** Over a billion people worldwide are infected, often for life, by a parasite known as *Toxoplasma gondii*. Most of the time, the parasites remain in a dormant state and cause no symptoms. However, the parasites can reactivate, which can be fatal for people with weaker immune systems. Currently no drugs can kill the dormant stages of *T. gondii*.

One of the possible reasons *T. gondii* is so widespread is because it easily passes from host to host, and resists both detection and treatment. Like many parasites, *T. gondii* infects multiple hosts during its life cycle: it reproduces sexually in its 'definitive' host, the cat, and reproduces asexually in its 'intermediate' hosts, which include virtually all warm-blooded animals. Unlike most parasites, all hosts can transmit *T. gondii* to other hosts, making it a very flexible parasite. However, its closest living relative, *Hammondia hammondi*, can only be transmitted from a definitive host to an intermediate host, or vice versa. The reason for this difference remains unclear, but comparisons between these species can hopefully help uncover how *T. gondii* came to be such a promiscuous parasite.

Now, Sokol, Primack *et al.* report that a dramatic difference between *H. hammondi* and *T. gondii* is the ability to adapt to changes in the environment. While *H. hammondi* grows in a very strict and inflexible way, *T. gondii* can sense its environment and change its growth rate accordingly. This is important because it would allow *T. gondii* to grow until the host immune response is activated, at which time *T. gondii* can then quickly become dormant and hide from these host defences. Sokol, Primack et al. suggest that this unique adaptation may have helped *T. gondii* to become the globally successful parasite that it is today, and explain why *H. hammondi* is much less lethal.

It is also possible that *T. gondii* reactivates after periods of dormancy because it senses changes in its environment. If so, and if scientists can identify the sensors required for this, it may also be possible to target them and effectively block reactivation in infected humans. Comparing *T. gondii* with *H. hammondi* may provide a good way to identify these sensors.

DOI: https://doi.org/10.7554/eLife.36491.002

*2003*), *H. hammondi* is known to infect rodents, goats, and roe deer (*Dubey and Sreekumar, 2003*; *Lorenzi et al., 2016*), and at least in mice it is naturally avirulent compared to *T. gondii*. Unlike *T. gondii*, *H. hammondi* has an obligately heteroxenous life cycle (*Dubey and Sreekumar, 2003*), where intermediate host stages (i.e., tissue cysts) are only capable of infecting the definitive host, and definitive host stages (i.e., oocysts) are limited to infection of an intermediate host.

In addition to these life cycle differences, these two parasite species have in vitro behaviors that mirror their in vivo differences. Specifically, when *T. gondii* sporozoites (VEG strain [*Jerome et al., 1998*]) are used to initiate infections in human host cells in vitro, they undergo a period of rapid replication, after which their growth rate slows and a subpopulation of parasites spontaneously convert to bradyzoite (e.g., cyst-like) stages (*Conde de Felipe et al., 2008*). However, *T. gondii* parasites cultivated in this manner still undergo multiple cycles of replication, egress and reinvasion, and indefinite propagation (*Jerome et al., 1998*). In contrast, when *H. hammondi* sporozoites are put into tissue culture they replicate slower than *T. gondii* (*Jerome et al., 1998*; *Sheffield et al., 1976*) and do not appear to be capable of subculture (either via passage of culture supernatants or after scraping and trypsinization; (*Sheffield et al., 1976*; *Riahi et al., 1995*). Ultimately in vitro cultivated *H. hammondi* form bradyzoite cyst-like stages that are infectious only to the definitive host (*Sheffield et al., 1976*; *Riahi et al., 1995*).

While the molecular determinants that control these dramatic phenotypic differences are unknown, the fact that life cycle restriction in *H. hammondi* occurs in vitro provides a unique opportunity to identify key aspects of *H. hammondi* biology that underlie its restrictive life cycle. By contrast, the life cycle restrictions in *H. hammondi* will allow for elucidation of the cellular and molecular differences in *T. gondii* that determine its comparatively flexible life cycle. It is important to note that with respect to other Apicomplexans (including *N. caninum* and *Plasmodium* spp.), it is the *T. gondii* life cycle that is atypical; the majority of coccidian Apicomplexans are obligately heteroxenous (*Dubey and Sreekumar, 2003*; *Dubey, 1998*; *Muller and Baker, 1982*). The most parsimonious explanation is that the obligately heteroxenous life cycle of *H. hammondi* is the ancestral state,

and that during speciation *T. gondii* acquired novel traits linked to any number of key parasite processes (regulation of development, stress sensitivity, interactions with the host environment) that underlie its life cycle flexibility.

The obligately heteroxenous life cycle of *H. hammondi* makes experimentally characterizing and manipulating this parasite species difficult in comparison to other closely related apicomplexan parasites. We have characterized the invasion and replication rates of *T. gondii* (VEG strain; [*Dubey, 2001*; *Speer et al., 1997*]) and *H. hammondi* (HhCatEth1 and HhCatAmer [*Lorenzi et al., 2016*; *Dubey et al., 2013*]) from sporozoite-initiated infections, representing the most thorough head-to-head comparison of the intermediate host life stages of these distinct parasite species. Using sporozoite derived infections, we have also compared the timing and frequency of in vitro differentiation and report the first ever transcriptome from replicating *H. hammondi*. Together, these data reveal previously unknown features of *H. hammondi* development. Using information gleaned from these experiments, we were able to successfully subculture *H. hammondi* both in vitro and in a murine model during a highly predictable window of infectivity and replicative capacity. Additionally, we generated the first ever transgenic *H. hammondi*, opening up the door to future molecular and genetic studies in this poorly understood parasite species. These data provide fundamental insight for understanding how virulence and pathogenicity have evolved in *T. gondii* through interspecies comparisons with its closest extant relative. Finally, we found that *H. hammondi* is resistant to alkaline pH-induced tissue cyst formation and determined that the *H. hammondi* transcriptome is refractory to alkaline pH exposure. Our data indicate that *H. hammondi* exists as a unique zoite life stage and follows a developmental path that is distinct from the canonical tachyzoite and bradyzoite life stages described for *T. gondii*. Moreover our work has identified sensitivity to external stressors as a likely culprit contributing to the intriguing developmental and virulence differences between *T. gondii* and *H. hammondi*, a finding that we speculate may have a direct impact on the ability of *T. gondii* to cause disease in immunocompromised humans and may also underlie its extensive host and geographic range.

## Results

### *T. gondii* replicates 2-3X faster than *H. hammondi*, but both species have equal invasion rates

While *T. gondii* oocysts and sporozoites have been studied extensively in vitro and in multiple hosts (*Jerome et al., 1998*; *Dubey, 2001*; *Speer et al., 1997*), *H. hammondi* and *T. gondii* have been directly compared in only a handful of studies (e.g., [*Dubey and Sreekumar, 2003*]). Therefore we performed a head-to-head characterization of sporozoite derived infections from both species during in vitro infections. As expected from previous work, we found that HhCatEth1 replicated significantly slower than TgVEG in multiple replicate assays (*Figure 1A,B*; ** $P$=<0.01, **** $P$=<0.0001). Based on median vacuole size (*Table 1*, *Figure 1A,B*), TgVEG replicated ~2–3X faster than HhCatEth1, with a maximum division time of 9.6 hr compared to 18 hr for HhCathEth1 (*Table 1*). Despite these differences in replication, TgVEG and HhCatEth1 sporozoites had similar infectivity rates in human foreskin fibroblasts (HFFs), defined as the number of vacuoles counted as a fraction of the number of input sporozoites (HhCatEth1 0.12 ± 0.02 N = 2; TgVEG; 0.10 ± 0.04 N = 2 p=0.64; *Figure 1C–D*). These data highlight parasite replication as a potentially important component in determining differences in virulence between these species, since pathogenesis in vivo is often (but not always) linked to parasite burden (*Saeij et al., 2005*).

### Both *T. gondii* and *H. hammondi* spontaneously form tissue cysts, but do so with different dynamics and efficiency

All *T. gondii* strains studied to date can be cultured indefinitely by serial passage in rodent intermediate hosts or in tissue/cell culture (*Croken et al., 2014a*) In contrast, when grown in tissue culture *H. hammondi* ultimately differentiates into a cyst form that is only orally infectious to the definitive feline host (*Dubey and Sreekumar, 2003*). Interestingly, some strains of *T. gondii* (e.g., strains VEG and EGS; [*Jerome et al., 1998*; *Dubey, 2001*; *Paredes-Santos et al., 2013*]) are known to spontaneously form cysts in tissue culture. To elucidate the temporal dynamics of differentiation and tissue cyst formation in *H. hammondi* and compare them to *T. gondii*, we infected HFFs with TgVEG or

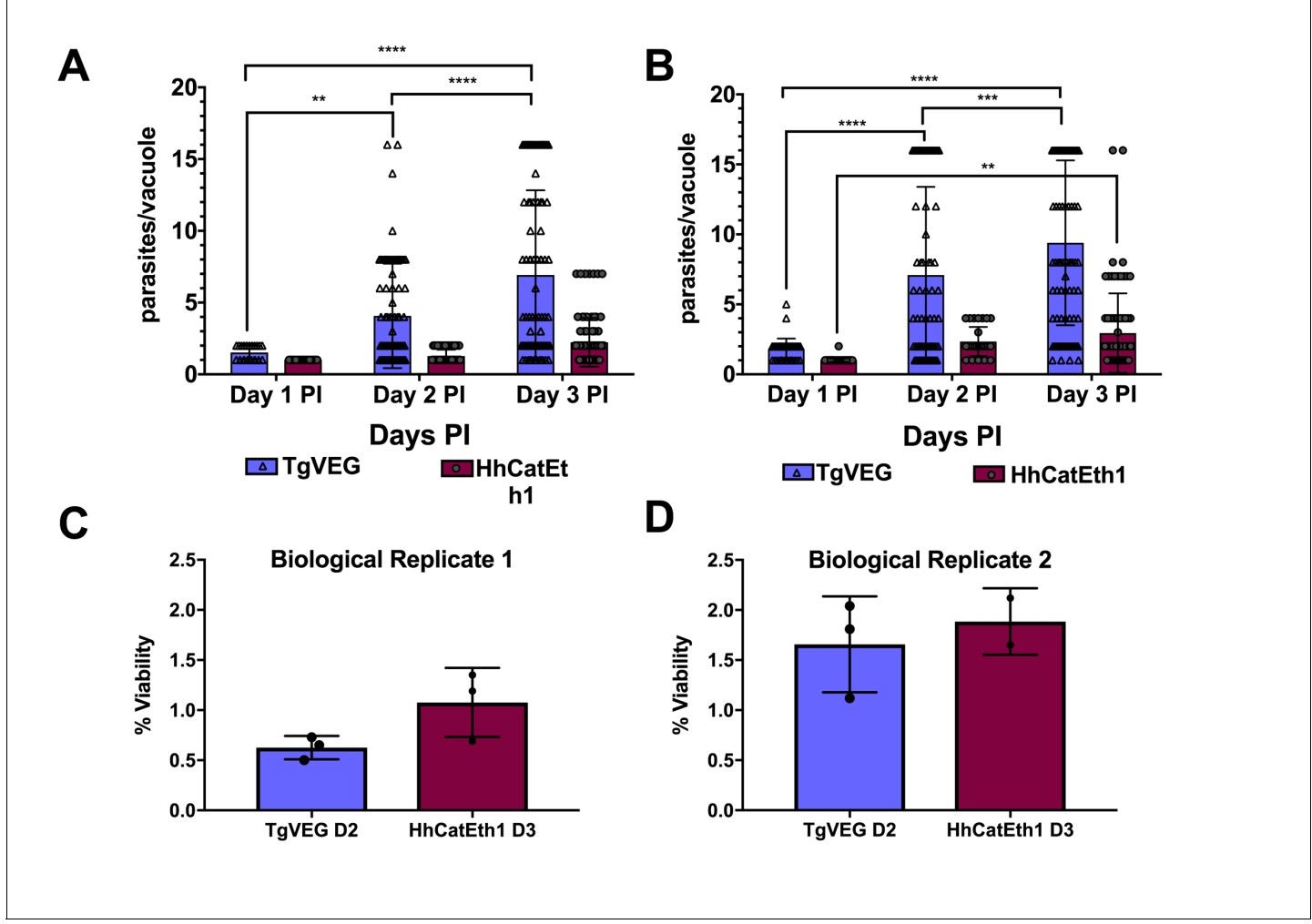

**Figure 1.** *T. gondii* and *H. hammondi* maintain similar infection rates, but *T. gondii* replicates at significantly higher rates. (A–B) Confluent monolayers of HFFs were infected with either HhCatEth1 or TgVEG sporozoites, obtained via in vitro excystation at an MOI of 0.5. Infected monolayers were fixed at 1, 2, and 3 DPI and assayed for the number of parasites observed in each vacuole for infection with *T. gondii and H. hammondi* for a total of 2 replicates, Replicate 1 (**A**) and Replicate 2 (**B**). Vacuoles containing more than 16 parasites were binned at 16 as it was defined as the limit of confident detection. Bars represent mean ± SD. For both Biological Replicate 1 and 2, there was a significant difference between the number of parasites per vacuole on Days 1, 2, and 3 PI for TgVEG. For Biological replicate 1, there was no significant difference between the number of parasites per vacuole for HhCatEth1 on Days 1, 2 and 3 PI. For Biological Replicate 2, there was a significant difference in the number of parasites per vacuole between 1 and 3 DPI for HhCatEth1. Statistical significance was determined with a 2-Way ANOVA and Tukey's multiple comparison test. (**C–D**) There is no significant difference in sporozoite infectivity between HhEthCat1 and TgVEG. Graphs demonstrate the in vitro percent viability of excysted HhCatEth1 and TgVEG sporozoites 2 and 3 days, respectively, post excystation. Bars show the mean ± SD of three technical replicates. The viability assay was repeated with two biological replicates and there was no significance in viability between species within experiments, determined via unpaired t-test with Welch's correction.

DOI: https://doi.org/10.7554/eLife.36491.003

HhCatEth1 sporozoites and stained them with *Dolichos biflorus* Agglutinin (DBA) at 4 and 15 days post-infection (DPI) to identify tissue cysts (***Figure 2A***). While all TgVEG and HhCatEth1 vacuoles were DBA-negative at 4 DPI, at 15 DPI ~80% (152/182) of *H. hammondi* vacuoles were DBA-positive, compared to 15% (34/355) of TgVEG vacuoles (p<0.0001; ***Figure 2B***). To more precisely determine the temporal dynamics of tissue cyst formation, we quantified spontaneous cyst formation with TgVEG (***Figure 2C***) or HhCatAmer (***Figure 2D***) over the course of 23 days. Both TgVEG and HhCatAmer spontaneously formed DBA-positive tissue cysts as early as 8 and 12 DPI, respectively. By 23 DPI 100% of all HhCatAmer vacuoles were DBA-positive, while the percentage of DBA-positive TgVEG vacuoles never exceeded 20% (***Figure 2C,D***). TgVEG cultures had to be passed three times

**Table 1.** Median and maximum parasites per vacuole during 1, 2, and 3 DPI for TgVEG and HhCatEth1.

| | Day 1 | Day 2 | Day 3 |
|---|---|---|---|
| | Median, Maximum | Median, Maximum | Median, Maximum |
| HhCatEth1 Replicate 1 | 1, 1 | 1, 2 | 2, 4 |
| HhCatEth1 Replicate 2 | 1, 2 | 2, 4 | 2, 16 |
| TgVEG Replicate 1 | 2, 2 | 2, 16 | 4, 16+ |
| TgVEG Replicate 2 | 2, 4 | 4, 16+ | 8, 16+ |

DOI: https://doi.org/10.7554/eLife.36491.004

over the course of the experiment to prevent complete lysis of the host cell monolayers, which is consistent with their propagation as a mixed population of tachyzoites and bradyzoites. These data demonstrate that *H. hammondi* undergoes a much more rigid developmental program during in vitro cultivation compared to *T. gondii*, resulting in 100% tissue cyst conversion and what is likely a near complete arrest in growth.

## The spontaneous differentiation process in *H. hammondi* is characterized by the increased abundance of hundreds of known bradyzoite-specific genes

Our previous work showed that the *H. hammondi* oocyst transcriptome (consisting of sporulated and unsporulated oocysts) was significantly more 'bradyzoite-like' than *T. gondii* sporozoites (*Walzer et al., 2014*), and we reasoned that this was consistent with the propensity of this species to spontaneously form cysts during in vitro cultivation. However, these transcriptome studies were from the quiescent oocyst life stage and provided no insight into the developmental dynamics of the *H. hammondi* transcriptome compared to *T. gondii*. To compare the dynamics of the replicating *H. hammondi* and *T. gondii* transcriptomes during in vitro development, we used RNAseq on day 4 (D4) and day 15 (D15) in vitro cultured *H. hammondi* and *T. gondii*. Using existing genome annotations (*Lorenzi et al., 2016*), we identified 7372 putatively orthologous genes between *T. gondii* and *H. hammondi* (ToxoDB.org; likely a conservative estimate of shared gene content between these species). As expected based on their in vitro growth rates (*Figure 1*), the number of reads mapping to the *H. hammondi* transcriptome was lower overall compared to those mapping to *T. gondii* (*Figure 3—figure supplement 1A*), and this was particularly pronounced at 15 DPI. To mitigate the effect this could have on falsely categorizing transcripts as being of higher abundance in *T. gondii* compared to *H. hammondi*, we included a given gene in downstream analyses only if one *H. hammondi* D4 and one *H. hammondi* D15 sample had at least five reads. This reduced the overall size of the queried gene set to 4146, and all analyses were performed with this subset.

When $\log_2$ transformed and DESeq2-normalized RPM (mapping Reads Per Million; all data in Supplementary Data File 1) values were plotted for D4 and D15 between species we observed a high correlation in transcript levels at both time points (*Figure 3—figure supplement 1B*). While the majority of genes were expressed at similar levels between *T. gondii* and *H. hammondi*, we identified multiple transcripts of significantly different abundance (*Figure 3A* and blue data points in *Figure 3—figure supplement 1B*).

We used Gene Set Enrichment Analysis (GSEA; [*Subramanian et al., 2005*]) with previously published (*Croken et al., 2014b*) and our own curated gene sets (*Table 2*) to identify gene classes with species-specific enrichment on D4 and D15 (see Methods and *Table 2* for Gene Sets). We identified seven gene sets that showed significant species-specific enrichment (FDR q-value <0.01; NES scores shown in *Figure 3B*). *H. hammondi* profiles showed significant enrichment for in vitro (D4, D15) and in vivo (D15) bradyzoite gene sets (*Figure 3B,C*; heat maps shown in *Figure 3—figure supplement 1D*), showing that replicating *H. hammondi* zoites maintain the 'bradyzoite-like' transcriptional profile previously observed in sporozoites after at least 15 days in culture (*Walzer et al., 2014*).

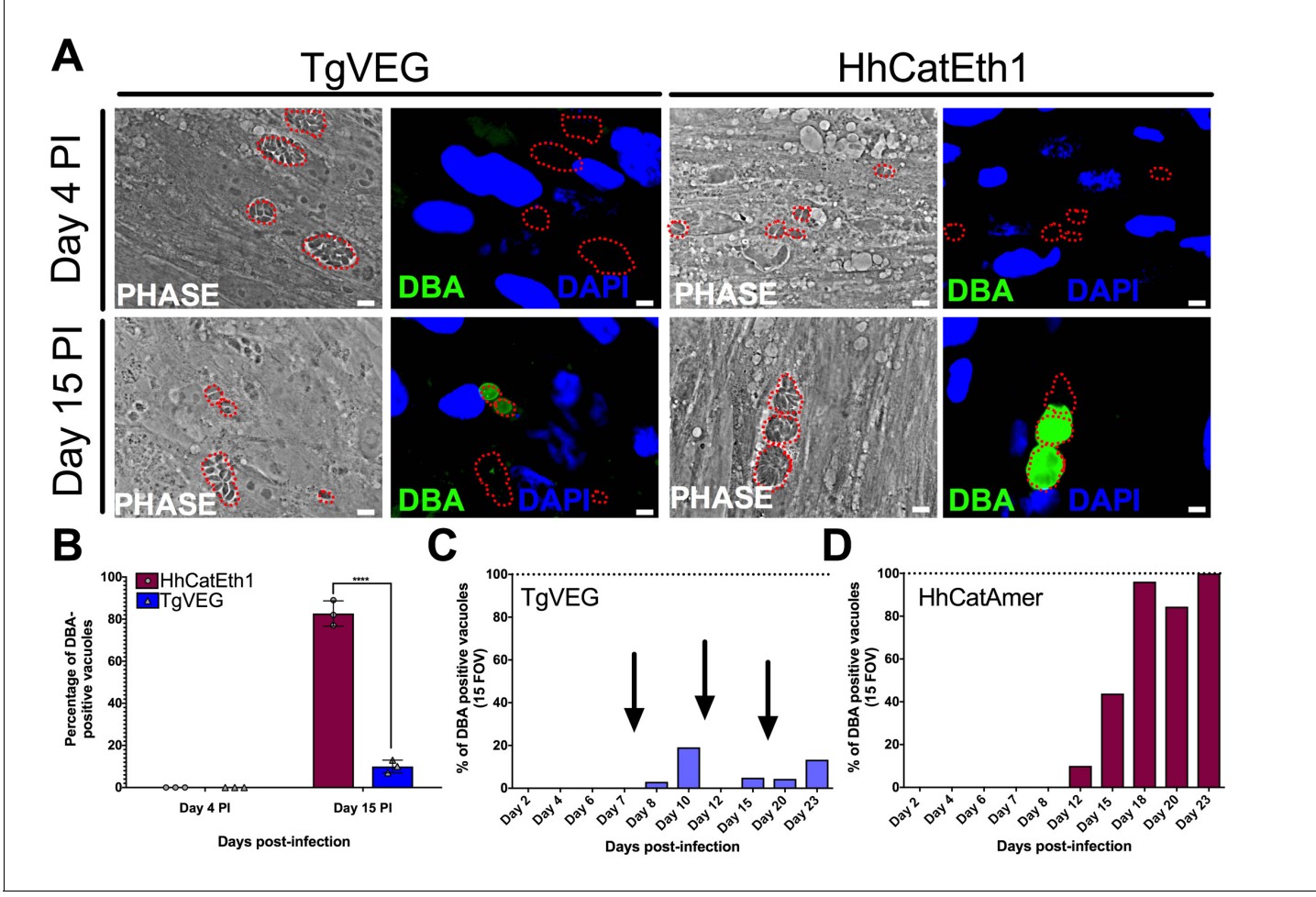

**Figure 2.** *T. gondii* and *H. hammondi* spontaneously form tissue cysts in vitro, but do so with different dynamics. (A) Representative images of cells infected with *T. gondii* (TgVEG) and *H. hammondi* (HhCatEth1) sporozoites that were fixed and stained with DBA after 4 and 15 DPI. Scale bar represents 5 μm. (B) Quantification of *T. gondii* and *H. hammondi* infection described in A demonstrating that after 4 days, neither *H. hammondi* or *T. gondii* spontaneously form tissue cysts, but both parasite species form tissue cysts after 15 DPI. However, the percentage of *H. hammondi* tissue cysts is significantly increased after 15 DPI when compared to *T. gondii*. Statistical significance was determined using a 2-way ANOVA with Sidak's multiple comparison test. (****=P < 0.0001). (C and D) Average percentage of DBA positive vacuoles in 15 FOV over a course of a 23 day infection with *T. gondii* (TgVEG) and *H. hammondi* (HhCatAmer). Arrows in (C) indicate when *T. gondii* was passed onto new host cells to prevent complete lysis. (D) *H. hammondi* form tissue cysts after 12 DPI. The percentage of DBA positive vacuoles increases until all vacuoles are DBA positive at 23 DPI. (C) *T. gondii* forms tissue cysts at 8 DPI, however, the percentage of DBA positive vacuoles does not reach 100% during the 23 day infection.

DOI: https://doi.org/10.7554/eLife.36491.005

Moreover, *T. gondii* showed significant enrichment compared to *H. hammondi* for in vitro tachyzoite genes at 15 DPI(*Figure 3B*; heatmap shown in *Figure 3—figure supplement 1C*), a time point at which VEG is <10% DBA-positive while *H. hammondi* is between 40–80% DBA-positive (*Figure 2B, C*). Overall these data are consistent with our general understanding of transcript abundance changes that occur during the tachyzoite to bradyzoite transition (albeit entirely from studies using *T. gondii*).

A more unexpected finding was that at both D4 and D15, *H. hammondi* had a transcriptional profile that was significantly enriched for genes of high transcript abundance in *T. gondii* merozoites replicating in cat enteroepithelial cells but poorly expressed in other life stages ([*Croken et al., 2014b*; *Behnke et al., 2014*]; CAT STAGE SPECIFIC 1, 2; *Figure 3C,D*). Specifically, out of the 21 detectable members of the 'CAT STAGE SPECIFIC 1' gene set, 18 were ranked in the top 308/4146 (7.4%) and 174/4146 (4.2%) of genes of greater abundance in *H. hammondi* D4 and D15 zoites, respectively (*Figure 3C*). To validate this finding we performed RNAseq on *T. gondii* and *H.*

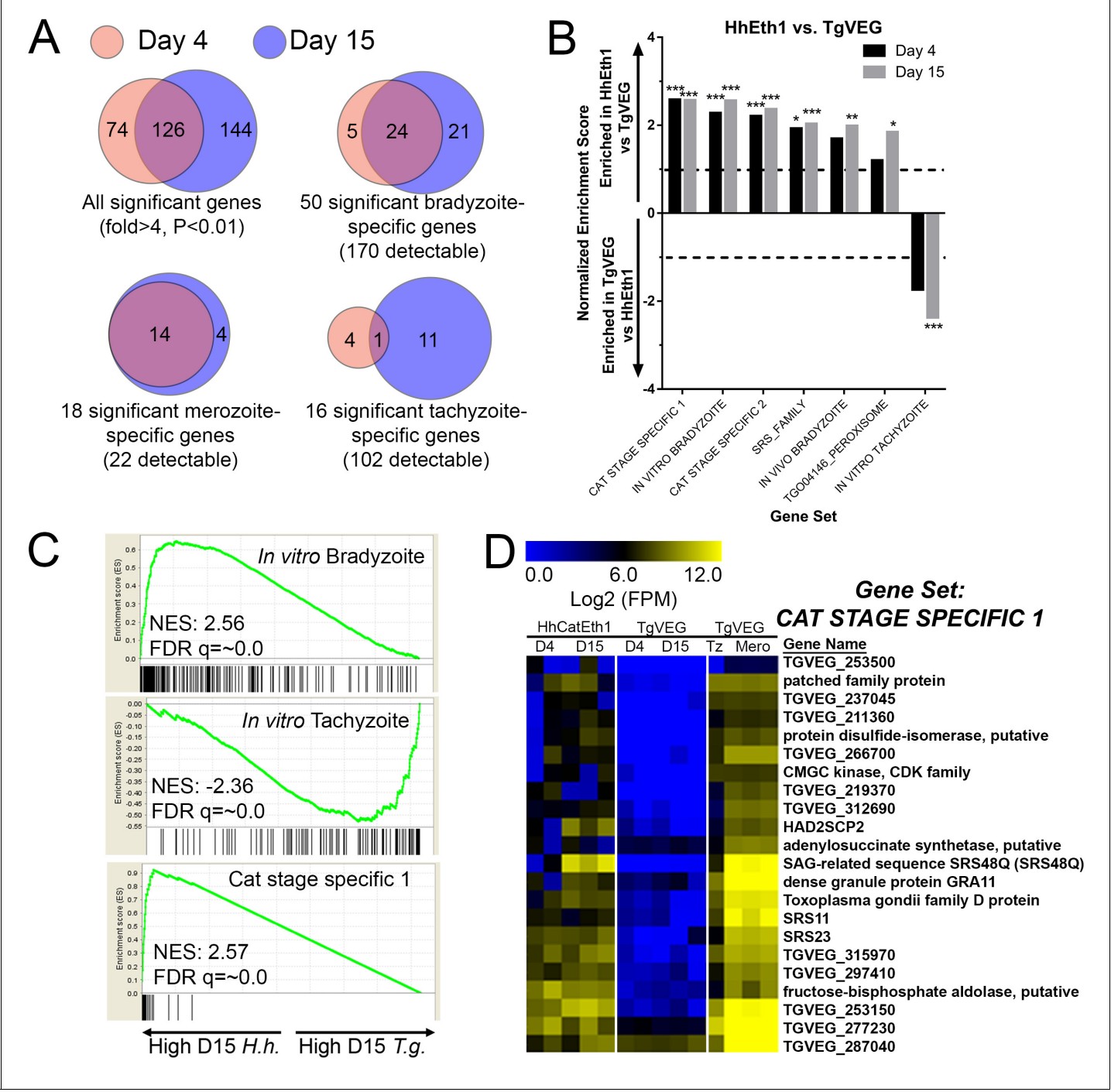

**Figure 3.** mRNAseq comparisons between *T. gondii* and *H. hammondi* identify unique aspects of the *H. hammondi* transcriptome. (A) Venn diagrams indicating the degree of overlap in genes of significantly different transcript abundance between *T. gondii* and *H. hammondi* at D4 and D15 post-infection. While overall there are distinct genes that are significantly different at D4 and D15 (Venn diagram, upper left), *H. hammondi* expresses high numbers of bradyzoite genes even at 4 dPI and this number increases further by D15 (21 additional genes; Venn diagram in upper right). This progression towards an increase in the number of bradyzoite genes over time is not recapitulated among merozoite-specific genes, in that only four additional detectable genes from this gene set were found to be of significantly higher abundance ($P_{adj}$ <0.01; Fold-change ≥4) at 15 DPI. Moreover 18 of the 22 total detectable merozoite genes were of significantly different abundance in *H. hammondi*, indicating its transcriptional similarity to both bradyzoites and merozoites. (B) Gene Sets found to be significantly (FDR q-value <0.05) enriched in *H. hammondi* high-abundance transcripts (top) or *T. gondii* high-abundance transcripts (bottom) at either 4 or 15 days in culture. Gene Set Details can be found in *Table 2*. *: p<0.05; **: p<0.01; ***: p<0.001. (C) GSEA plots of the in vitro bradyzoite, in vitro tachyzoite, and Cat stage specific one gene sets, showing enrichment profiles between *H.*

*Figure 3 continued on next page*

*Figure 3 continued*

*hammondi* and *T. gondii* at 15 DPI. NES; Normalized enrichment score. FDR q: False discovery rate q-value. (**D**) Hierarchically clustered expression data for detectable genes in the CAT STAGE SPECIFIC one gene set in *H. hammondi* or *T. gondii* grown for 4 or 15 days in vitro, or for *T. gondii* grown as tachyzoites or merozoites (in cat enteroepithelial cells).

DOI: https://doi.org/10.7554/eLife.36491.006

The following figure supplements are available for figure 3:

**Figure supplement 1.** Transcriptional profiling and Gene Set Enrichment Analysis of *T. gondii* and *H. hammondi*.

DOI: https://doi.org/10.7554/eLife.36491.007

**Figure supplement 2.** mRNA-seq comparisons between TgVEG and HhCatAmer sporozoites identify unique *H. hammondi* transcriptome profiles.

DOI: https://doi.org/10.7554/eLife.36491.008

**Figure supplement 3.** qPCR validation for nine transcripts that were found to be of higher abundance in D4 or D15 *H. hammondi* compared to *T. gondii*.

DOI: https://doi.org/10.7554/eLife.36491.009

*hammondi*-infected THP-1 cells 24 hr post-infection and found a consistent and highly significant enrichment for these same gene sets (*Figure 3—figure supplement 2A–C*), and in this case a similar number of reads from *H. hammondi* and *T. gondii* mapped to their respective parasite transcriptomes (*Figure 3—figure supplement 2D*). These data suggest that *H. hammondi* has aspects of its transcriptional profile that are independent of its spontaneous conversion to bradyzoites and that a subset of genes that are uniquely expressed in *T. gondii* merozoites and sporozoites may not share these same profiles in *H. hammondi*.

We also validated a subset of transcripts that were of greater abundance in D4 and D15 cultivated *H. hammondi* parasites using qPCR (heatmap of selected genes in *Figure 3—figure supplement 3A*). We used *ROP18* as a control for a gene that was of significantly higher abundance in *H. hammondi* compared to *T. gondii* (TgVEG and other Type III strains express very little *ROP18* transcript due to a promoter indel; [*Walzer et al., 2013*; *Walzer et al., 2014*; *Boyle et al., 2008*]), and *CDPK1* and *AMA1* as control genes of similar abundance between the two species. *GRA1* transcript levels were used as loading controls. We used validated primer sets for eight genes (which had to be made separately for each species; see *Supplementary file 3* for all primer sequences) to quantify eight transcripts (plus *ROP18*) using qPCR on RNA freshly isolated from D4 and D15 HhCatEth1 and TgVEG cultures. As expected *ROP18* transcript was found to be >1000 fold greater in abundance in HhCatEth1 compared to TgVEG, while *CDPK1* and *AMA1* transcript levels were not significantly different between species (*Figure 3—figure supplement 3B*). The remaining nine queried transcripts showed significantly higher abundance in *H. hammondi* compared to *T. gondii* (*Figure 3—figure supplement 3B*) confirming the validity of our approach to analyze, normalize and filter cross-species RNAseq data with significant differences in read counts.

## *H. hammondi* can be subcultured successfully in vitro

From a phylogenetic standpoint *H. hammondi* is a good model to understand the genetic basis for traits that are unique to *T. gondii*. However, unlike all *T. gondii* strains studied to date, sporozoite-initiated cultures of *H. hammondi* in host cells from multiple organisms including humans, cattle, and non-human primates lose their ability to be subcultured in vitro (*Sheffield et al., 1976*; *Riahi et al., 1995*). In order to precisely define when *H. hammondi* sporozoites lose the ability to infect new cells

**Table 2.** Additional gene sets used in Gene Set Enrichment analyses.

| Gene set name | Search strategy on ToxoDB | Size |
| --- | --- | --- |
| SRS_Family | Text search for 'SAG-related sequence SRS' | 111 |
| AP2_family | Text search for 'AP2 domain transcription factor' | 66 |
| CAT STAGE SPECIFIC 1 | Cat enteroepithelial stages at least 40-fold higher than Tachyzoite at D3, D5 or D7 | 293 |
| CAT STAGE SPECIFIC 2 | Cat enteroepithelial stages > 90 th percentile in any of D3, D5 or D7; Tachyzoite expression between 0 and 50th percentile | 81 |

DOI: https://doi.org/10.7554/eLife.36491.010

we attempted to subculture *H. hammondi* zoites after defined periods of cultivation (*Figure 4A–B*). We found that *H. hammondi* zoites were capable of infecting and replicating within new host cells for up to 8 days post-excystation (*Figure 4C–F*), while we did not observe any evidence of growth when subcultures were initiated on days 11 and 15 post-excystation (*Figure 4G,H*). In agreement with previous studies (*Dubey and Sreekumar, 2003*; *Conde de Felipe et al., 2008*; *Sheffield et al., 1976*) we also observed that the number of visible *H. hammondi* vacuoles disappear over time

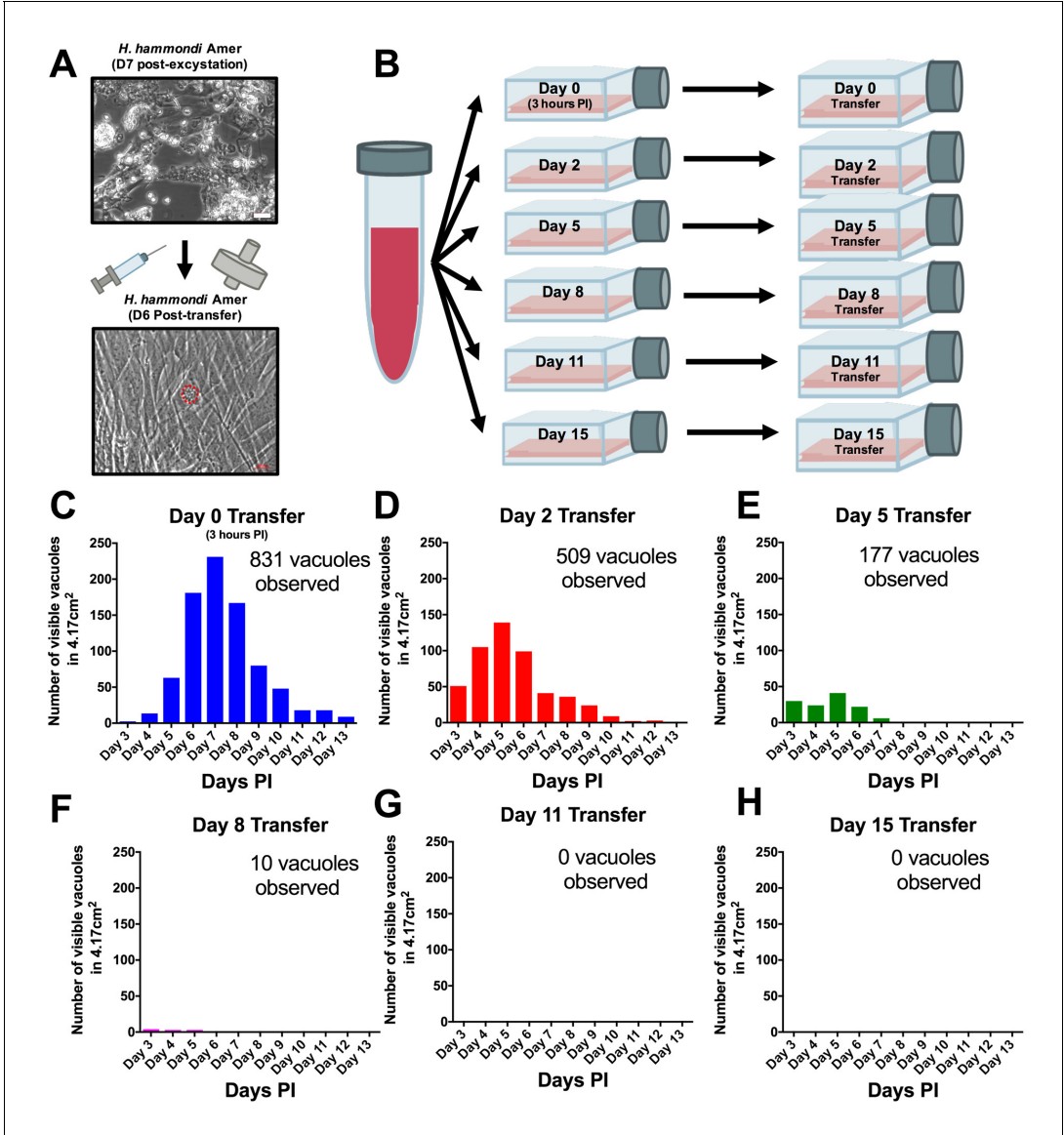

**Figure 4.** *H. hammondi* can be successfully subcultured in vitro for a limited period of time. (**A**) Monolayers containing HhCatAmer with oocyst debris were needle passaged, filtered, and used to infect a confluent monolayer of HFFs seven days post-excystation. Vacuoles were observed in subcultured monolayers after 6 DPI. (**B–H**) Confluent monolayers of HFFs were infected with HhCatAmer sporozoites at an MOI of ~2. (2,812,500 sporozoites). After a three-hour incubation at 37°C, the Day 0 transfer flask was scraped, needle passaged, filtered, and transferred to a new host cells. This process was repeated after 2, 5, 8, and 15 days of growth. Each flask was monitored daily for the number (**C–H**) for 13 DPI, or until visible vacuoles were no longer detected. The number of visible vacuoles increased in number for the first week post-excystation then began to decrease. Replicative capacity was greatest for sporozoites transferred to new host cells at Day 0 (**C**) and decreased during each subsequent passage. (**D–H**).
DOI: https://doi.org/10.7554/eLife.36491.011

The following figure supplement is available for figure 4:

**Figure supplement 1.** *H. hammondi* vacuole sizes following subculture.
DOI: https://doi.org/10.7554/eLife.36491.012

(*Figure 2C–E*). We did not identify any vacuoles derived from lysing and re-invading parasites (which would have comparatively smaller sizes) after 10 DPI as we only observed a general increase in vacuole size prior to their disappearance from culture (*Figure 4—figure supplement 1*). Together, our data support previous work showing that *H. hammondi* cannot be subcultured with high efficiency after approximately one week or longer in culture (*Sheffield et al., 1976*; *Riahi et al., 1995*), but importantly we identified a previously unknown window prior to spontaneous tissue cyst formation where this parasite can be effectively subcultured.

When we performed similar experiments in vivo, we observed that the ability to be subcultured in vitro and infectivity in a mouse model of infection were temporally correlated. We cultivated *H. hammondi* sporozoites in HFFs for 5 days and used them to infect mice intraperitoneally (50,000 zoites). As early as 2 DPI, we measured interferon-gamma (IFNγ) production indicative of an active infection (*Figure 5A*). We used PCR to detect *H. hammondi* DNA in spleens and resident peritoneal cells in these same mice 3 weeks post-infection (PI) (*Figure 5B*). Histological examination of muscle from infected mice using anti-*Toxoplasma* antibodies that cross-react with *H. hammondi* showed the presence of the parasites in the muscle tissue of the mouse infected with in vitro-passed *H. hammondi* (right panel, *Figure 5C*) which were similar to those observed after oral inoculation with *H. hammondi* sporulated oocysts (middle panel; *Figure 5C*). Overall these data show that *H. hammondi* parasites grown in vitro are capable of (a) inducing an immune response in mice during the acute phase of infection, and (b) disseminating to multiple mouse tissues and establishing a chronic infection that resembles an infection resulting from oral gavage with sporulated oocysts. To determine how long mouse infectivity was retained in in vitro cultivated *H. hammondi*, mice were injected intraperitoneally with 50,000 zoites grown in vitro for 6, 8, 10, 13, and 15 days, and monitored for signs of infection by measuring serum IFNγ levels. The ability to be subcultured in vitro correlated perfectly with the ability to initiate infections in mice in vivo. Specifically, we observed an increase in IFNγ levels in mice infected with 6 DPI or 8 DPI in vitro zoites (*Figure 5D*), while we did not see an increase in IFNγ levels in those infected with parasites cultured for 10, 13, and 15 days (*Figure 5D*). To our knowledge this is the first time that sporozoite-derived growth stages of *H. hammondi* have been used to initiate infections in an intermediate host, since all previous studies focused on parasite that had already terminally differentiated.

## Generation of transgenic *Hammondia hammondi*

Genetic manipulation of *H. hammondi* has not been reported in the literature. Given the genetic similarity between *T. gondii* and *H. hammondi* (*Walzer et al., 2013*; *Lorenzi et al., 2016*) and the apparent interchangeability of promoter sequences between them (*Walzer et al., 2013*; *Walzer et al., 2014*), we transfected ~4 million HhCatAmer sporozoites with a UPRT-targeting CRISPR/CAS9-GFP plasmid (*Shen et al., 2014*) along with a PCR2.1 Topo-cloned *dsRED* cassette with 20 bp sequences flanking the CAS9 nuclease cut site within the *H. hammondi* UPRT gene. We used the *Toxoplasma*-specific plasmid to target *H. hammondi* UPRT as the gRNA sequence is identical between *T. gondii* and *H. hammondi*. Both plasmids were taken up and expressed by *H. hammondi*, as evidenced by GFP-fluorescence in the parasite nuclei and red fluorescence in the cytoplasm (*Figure 6A*). Importantly, transgenic parasites replicated in vitro since we observed multicellular vacuoles in the infected monolayer (*Figure 6A*). Transfection efficiency of viable parasites determined by counting the percentage of replicating parasites with CAS9-GFP and/or dsRED fluorescence was ~4% (15 out of 350 vacuoles at 48 hr post-transfection), which is consistent with typical transfection efficiencies observed with most strains of *T. gondii* (*Kim and Weiss, 2004*).

To make a stable transgenic parasite line, we transfected ~4 million parasites with the Hh*UPRT*:dsRED repair cassette and the Hh*UPRT*gRNA:CAS9:GFP plasmid and cultured them in 10 μM FUDR for 5 days. After FACS sorting ~12,000 dsRED-expressing zoites (~3% of the total parasite population; *Figure 6B*), we injected two female Balb/c mice intraperitoneally with ~6000 dsRED-expressing zoites each. For the first week of the infection, mice were injected 1X daily with 200 μL of 1 mM FUDR, and mice were euthanized and fed to a cat 4 weeks PI. Fecal floats showed the presence of dsRED-expressing sporozoites within some of the oocysts (*Figure 6B*), and ~32% of isolated sporozoites expressed dsRED based on FACS analysis (*Figure 6B*, bottom FACS plot). PCR screening (*Figure 6—figure supplement 1*) and the inability to grow parasites long term in vitro confirmed the identity of the fluorescent parasites as *H. hammondi*.

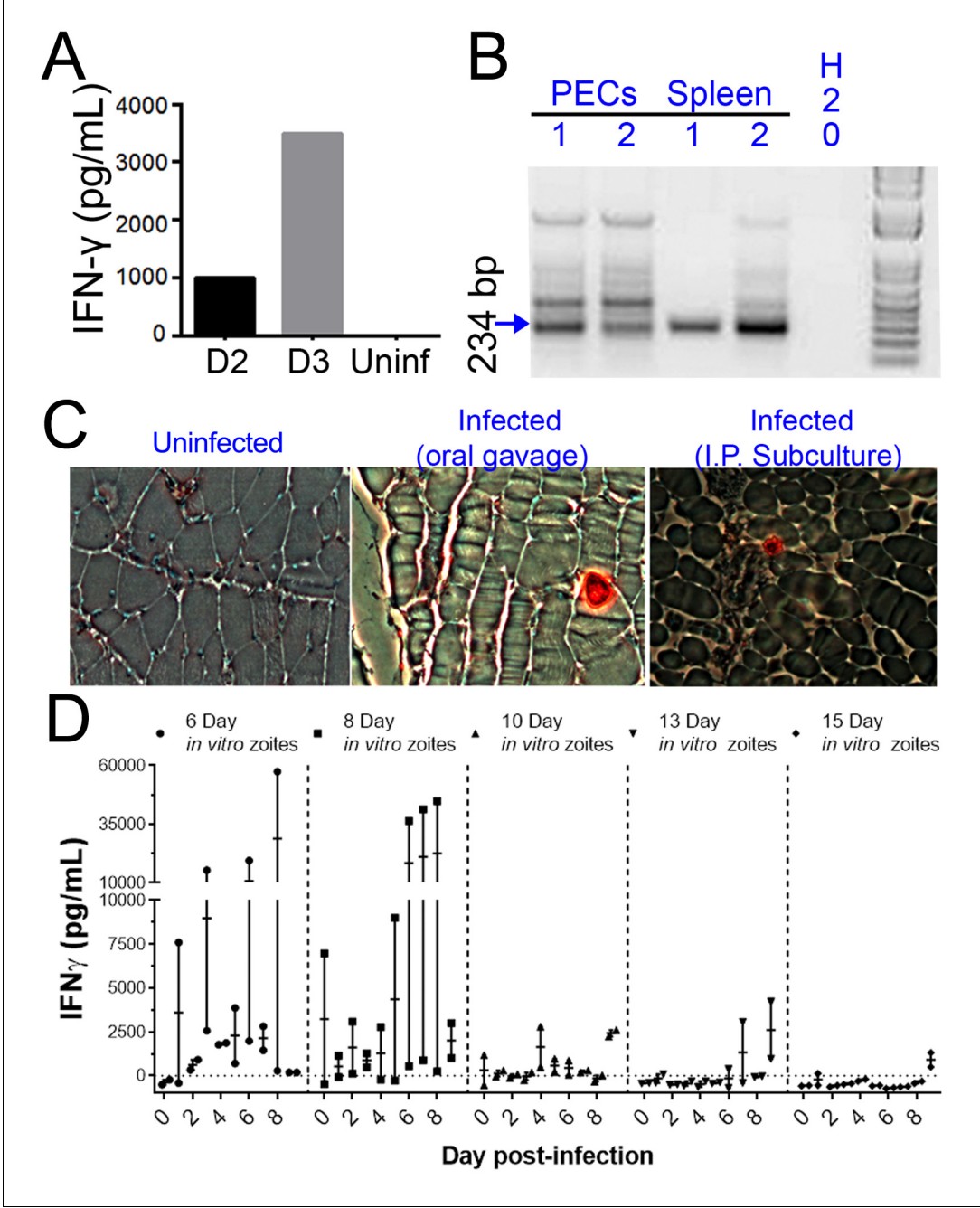

**Figure 5.** *H. hammondi* parasites can be grown in vitro and then used to successfully infect mice. (**A–C**) Two Balb/C mice were infected with ~50,000 *hr. hammondi* zoites grown in vitro for 5 days. (**A**) Serum was collected on days 2 and 3 and assayed for interferon-γ. (**B**) Three weeks post-infection, DNA was harvested from peritoneal lavage cells and spleen and assayed for *H. hammondi* DNA using *H. hammondi*-specific primers. Arrow indicates primary band for *H. hammondi*-specific PCR, although we also routinely see larger bands after PCR amplification. Samples were only considered positive if the 234 bp band was present (arrowhead). (**C**) Leg muscles from infected mice were sectioned and stained with *H. hammondi*-reactive anti-*Toxoplasma* antibodies and compared to uninfected mice and mice infected with *H. hammondi* by oral gavage with 50,000 sporulated oocysts. (**D**) In vitro cultivation leads to a dramatic and predictable loss in the ability to infect mice. Two BALB/C mice were infected with 50,000 *H. hammondi* American parasites after 6, 8, 10, 13, and 15 days of in vitro growth. The mass of the mice was monitored and a serum sample was obtained daily for 9 days post-infection. Analysis of IFN-γ production was analyzed in serum samples using an ELISA. Mice infected with parasites grown in vitro for 6 days (1 of 2) showed

*Figure 5 continued on next page*

*Figure 5 continued*
spikes in IFN-γ levels. This IFN-γ spike was also observed in mice infected with parasites grown in vitro for 8 days (2 of 2), while no IFN-γ was observed in mice infected with parasite grown in vitro for 10, 13, or 15 days.
DOI: https://doi.org/10.7554/eLife.36491.013

To determine if our transgenic approach to disrupt the *UPRT* gene and insert a dsRED expression cassette was successful, we quantified the growth of dsRED$^+$ and dsRED$^{(-)}$ parasites in the presence of FUDR and compared it to wild type HhCatAmer. Fixed and permeabilized parasites were stained with cross-reactive anti-*Toxoplasma* antibodies to identify all vacuoles. After 4 DPI, we quantified vacuole size and dsRED-expression status in >100 vacuoles for each strain (WT and dsRED) and condition (Vehicle or 20 μM FUDR). As expected, HhCatAmer:WT grew poorly in FUDR (*Figure 6C*), while HhCatAmer:dsRED parasites grew similarly to untreated parasites (*Figure 6C*). When we

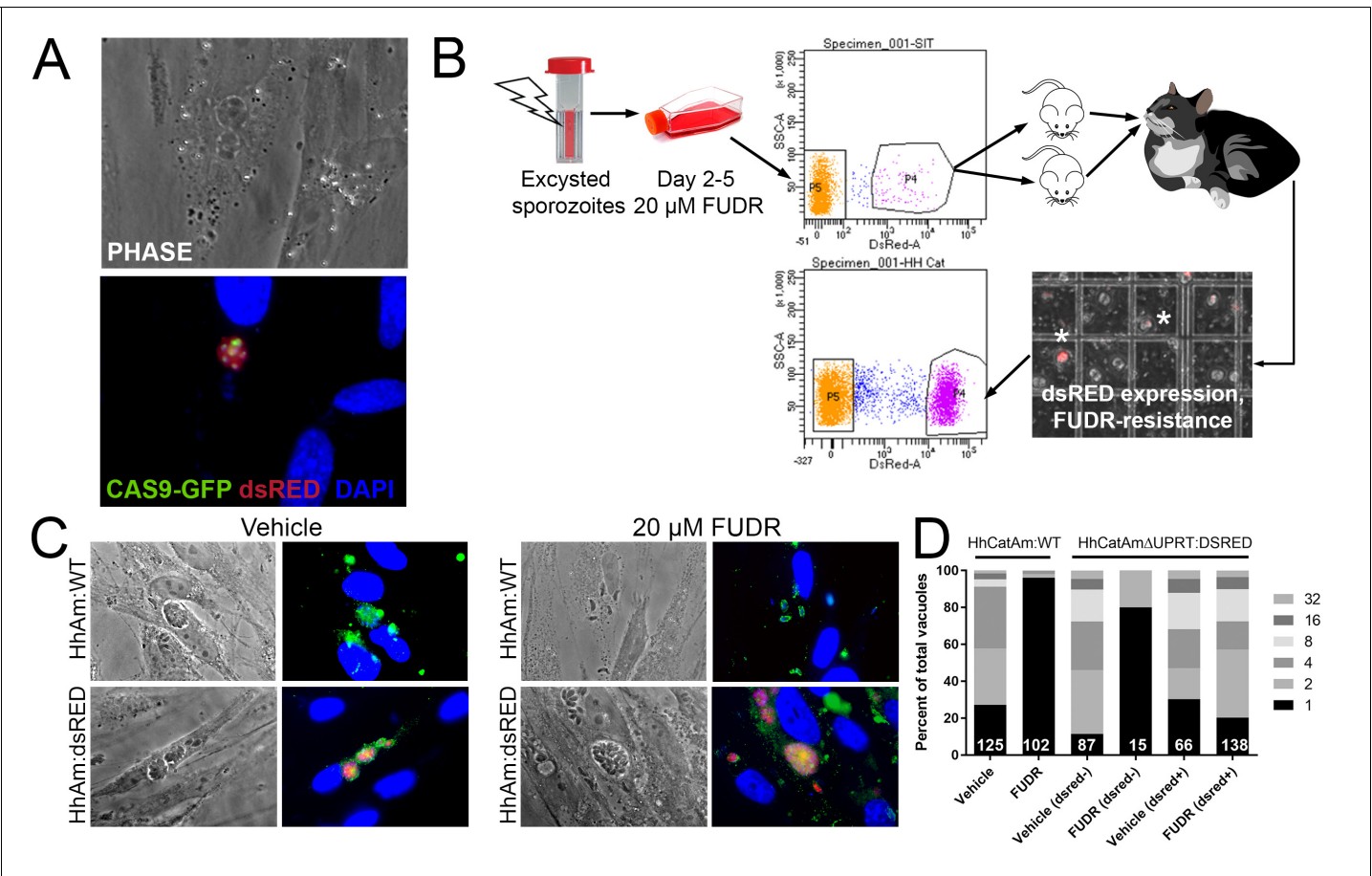

**Figure 6.** Generation of stable transgenic *Hammondia hammondi*. (**A**) *H. hammondi* zoites co-transfected with CRISPR/CAS9-GFP and a plasmid harboring a *T. gondii* dsRED expression cassette. We identified multiple parasites with both GFP-tagged nuclei (due to CAS9-GFP expression) and dsRED-tagged cytoplasm, indicating uptake of both plasmids within the same parasites. (**B**) Protocol for generating stably transgenic *H. hammondi* using flow cytometry and drug selection to enrich for transgenic parasites prior to cat infections. (**C,D**) Testing FUDR resistance in transgenic *H. hammondi* parasites. WT and transgenic *H. hammondi* sporozoites were incubated in the presence or absence of 20 μM FUDR, and vacuole size and presence/absence of dsRED was quantified in at least 100 vacuoles per treatment condition. While WT and non-dsRED-expressing *H. hammondi* from the transgenic population were highly susceptible to FUDR treatment (the majority of vacuoles in these categories contained only one parasite), dsRED-expressing transgenic parasites were resistant to FUDR treatment, confirming that the parasites harbored two genetic markers.
DOI: https://doi.org/10.7554/eLife.36491.014

The following figure supplement is available for figure 6:

**Figure supplement 1.** Validation of transgenic *H. hammondi*.
DOI: https://doi.org/10.7554/eLife.36491.015

separated parasite growth in FUDR based on dsRED expression, we found that dsRED$^{(-)}$ parasites in the *H. hammondi* population were just as susceptible to FUDR as were HhCatAmer:WT parasites (***Figure 6D***, first four bars), while dsRED$^{+}$ parasites grew equally well in FUDR compared to Vehicle (***Figure 6D***, right two bars). These data suggest that most, if not all, of the dsRED-expressing parasites are also null for the *UPRT* gene, either by direct replacement of the *UPRT* gene with the dsRED cassette or by incorrect non-homologous end joining and insertion of the dsRED cassette at another genomic location. This represents the first stably transgenic *H. hammondi* line. Furthermore, PCR analysis of linearly amplified genomic DNA from the HhCatAmer population expressing dsRED demonstrated that the Hh*UPRT*:dsRED repair cassette was incorporated into the genomic DNA of parasite within this population (***Figure 6—figure supplement 1***).

## *H. hammondi* is completely resistant to pH-induced cyst formation

While *T. gondii* can spontaneously form tissue cysts following an in vitro sporozoite infection, it can also be chemically induced to form bradyzoites using a variety of treatments (***Jerome et al., 1998***). Given the dramatic developmental differences between *T. gondii* and *H. hammondi* in vitro and in vivo, we reasoned that these parasite species may respond differently to external stressors that induced cyst formation. We therefore exposed *T. gondii* and *H. hammondi* zoites to alkaline pH bradyzoite induction media and identified cysts using DBA staining as above. As expected we found that TgVEG vacuoles grown in bradyzoite induction media were ~55% DBA-positive after 2 days of exposure and ~90% positive after 4 days of alkaline pH treatment (***Figure 7A,B***). However, we did not find a single DBA-positive *H. hammondi* vacuole at any time point (out of 297 total vacuoles; ***Figure 7A–B***). Given the importance of parasite replication events and bradyzoite conversion (***Radke et al., 2003***), it is possible that the lack of cyst conversion in *H. hammondi* over a 4 day treatment was due to fewer replication cycles. While vacuole sizes for *T. gondii* and *H. hammondi* differed significantly after 2 days of growth in standard and bradyzoite induction media (***Figure 7C***), pH treatment had no significant effect on vacuole size in either species (***Figure 7C***; p=0.63, 1.0, respectively). This result suggests that bradyzoite conversion occurs independently of changes or differences in replication rate and provides strong support for the fact that *H. hammondi* are immune to cellular stresses induced by bradyzoite induction conditions.

## The *H. hammondi* transcriptome is mostly non-responsive to high pH exposure

Given the well-established impact of alkaline pH media on *T. gondii* cyst formation in vitro and the propensity of *H. hammondi* to spontaneously form cysts in vitro, we found it remarkable that we did not identify a single DBA-positive *H. hammondi* cyst after even 4 days of alkaline pH exposure. An intriguing hypothesis emerging from this result is that *T. gondii* is more adept than *H. hammondi* at responding to stress signals in its environment. To test this hypothesis with respect to pH exposure, we treated *T. gondii* and *H. hammondi* sporozoites with bradyzoite induction media for 2 days and performed RNAseq analysis to identify changes in their respective transcriptomes (***Figure 8—figure supplement 1A***). For *T. gondii* treated with alkaline pH, we observed significant enrichment in the IN VITRO BRADYZOITE and IN VIVO BRADYZOITE (***Croken et al., 2014b***) gene sets (NES = −2.4902,–1.800, respectively; FDR q =~0.00 for both) for TgVEG treated with alkaline pH (***Figure 8A,B***), and the IN VITRO TACHYZOITE and IN VIVO TACHYZOITE gene sets under control conditions (NES = 1.94, FDR q = 0.001; NES = 1.87, FDR q = 0.002, respectively; ***Figure 8A***). This was in stark contrast to HhCatAmer in which members of the IN VITRO BRADYZOITE gene set showed no enrichment after alkaline pH exposure (NES = −1.1681, FDR q = 0.2078; ***Figure 8A–B***). For the *H. hammondi* transcriptome, the IN VITRO BRADYZOITE gene set seemed to consist of two classes of genes: (a) those that were not induced upon pH exposure in *H. hammondi* but were in *T. gondii* (top half of heatmap in ***Figure 8C***) and (b) those that were of high abundance in *H. hammondi* prior to exposure to high pH (lower half of heatmap in ***Figure 8C***). While we identified 90 transcripts that significantly differed in TgVEG following pH exposure (fold change value $\geq$2 and P$_{adj}$ value <0.01), we only identified 24 *hr. hammondi* transcripts with pH-altered abundance (these are shown in ***Figure 8D***). Importantly while the majority of pH-altered transcripts were of higher abundance in *T. gondii*, 21 out of 24 pH-altered transcripts in *H. hammondi* were of lower abundance, indicating that *H. hammondi* and *T. gondii* have distinct responses to high pH bradyzoite induction

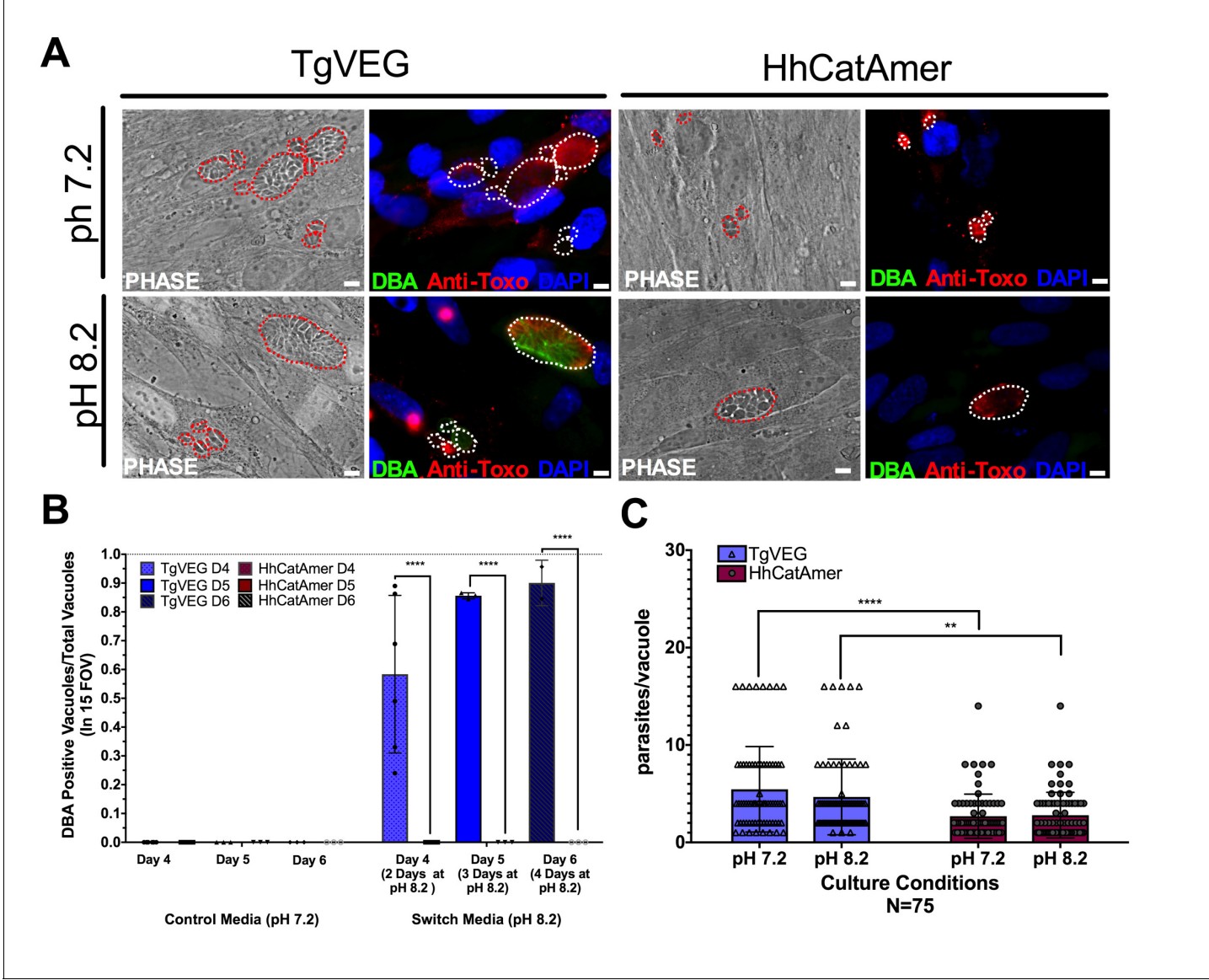

**Figure 7.** Conditions that induced *T. gondii* tissue cyst formation in vitro did not induce tissue cyst formation in *H. hammondi*. (**A**) Representative images of cells infected with *T. gondii* (TgVEG) and *H. hammondi* (HhCatAmer) sporozoites that were grown for 2 days in pH 7.2 media and switched to pH 8.2 media (bradyzoite induction conditions) for 2 days, fixed and stained with an Anti-*Toxoplasma gondii/Hammondia hammondi* antibody and DBA. Scale bar represents 5 μm. (**B**) Quantification of *T. gondii* and *H. hammondi* tissue cyst formation described in A demonstrating that *H. hammondi* does not form tissue cysts when exposed to conditions that promote tissue cyst formation in *T. gondii*. Statistical significance was determined using a 2-way ANOVA with Tukey's multiple comparison test. (****=P < 0.0001). (**C**) Number of parasites per vacuole for both *T. gondii* and *H. hammondi* grown at pH 7.2 and pH 8.2. *T. gondii* has significantly more parasites per vacuole than *H. hammondi* at both pH 7.2 and pH 8.2. There is no statistically significant difference between either *T. gondii* or *H. hammondi* between pH 7.2 and pH 8.2. Statistical significance was determined using a 2-way ANOVA with Sidak's multiple comparison test. (**=P < 0.01 and ****=P < 0.0001).

DOI: https://doi.org/10.7554/eLife.36491.016

(*Figure 8C,D*; See *supplementary file 2* for normalized RNAseq data). Since transcripts for the canonical bradyzoite genes *BAG1*, *LDH2*, and *ENO1* (*Weiss and Kim, 2000*; *Lyons et al., 2002*) were below the levels of detection defined for *H. hammondi* samples in our RNAseq analysis (see Materials and methods), we conducted qPCR analysis of these genes in TgVEG and HhCatAmer treated with alkaline pH for 2 (TgVEG only) and 3 days. We found a significant increase in transcript abundance for all three genes in TgVEG exposed to pH 8.2 for 2 days (*BAG1* p=0.0023, *ENO1* p=0.0047, *LDH2* p=0.0079). In HhCatAmer, we did observe a significant pH-dependent increase in

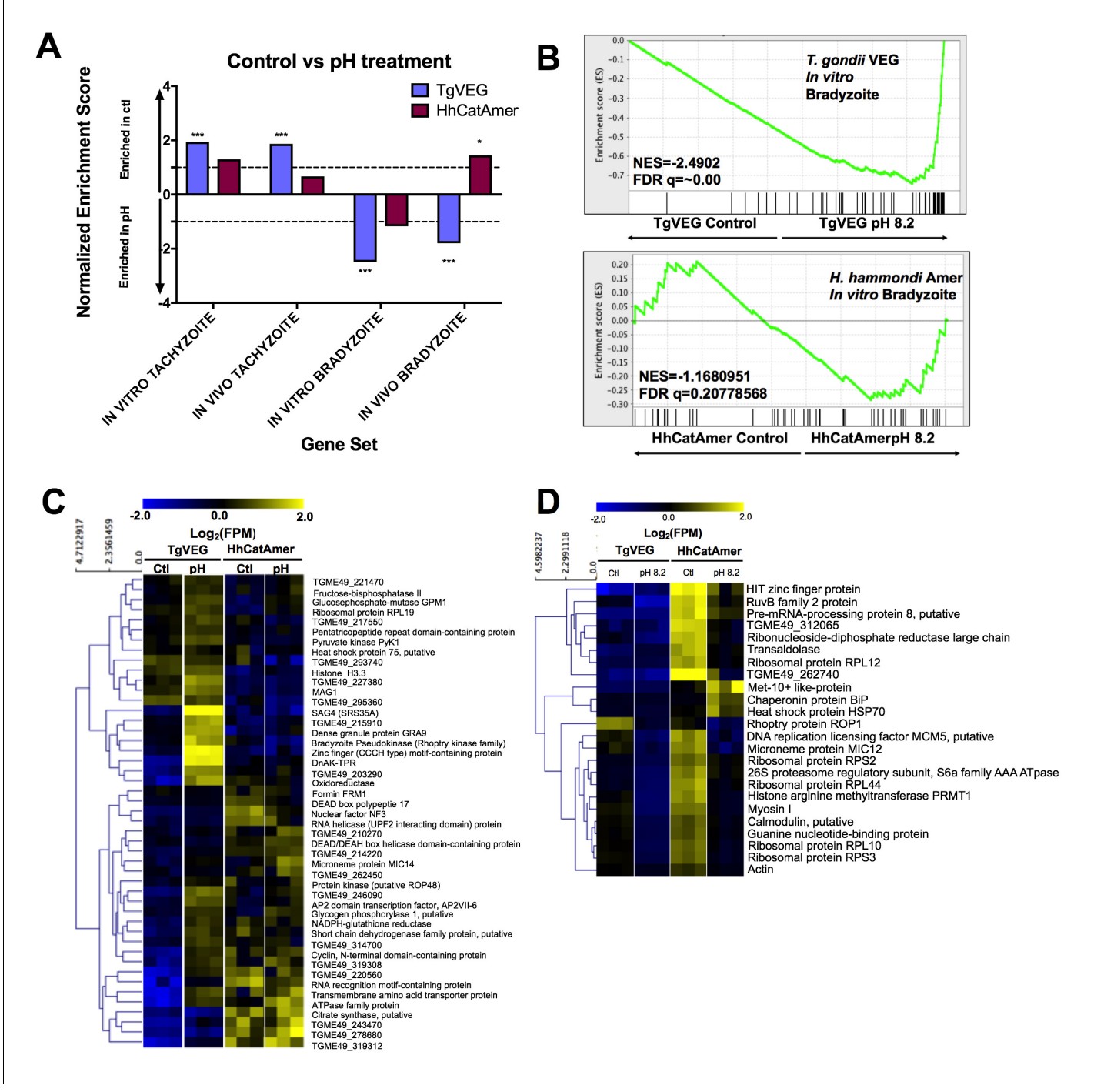

**Figure 8.** The *H. hammondi* transcriptome is refractory to pH-induced stress. (**A**) GSEA analysis comparing TgVEG (blue) and HhCatAmer (red) treated with control media to TgVEG and HhCatAmer treated with pH 8.2 media. Not all gene sets are significantly enriched. NES: Normalized enrichment score FDR q: False discovery rate q-value. See table supplement for FDR q-values for all analyses. Nominal p-value is indicated by *<0.05, **<0.01, and ***<0.001. (**B**) GSEA plots of the IN VITRO BRADYZOITE gene set comparing control versus pH 8.2 treated TgVEG and HhCatAmer. (**C**) Heat map showing mean centered log$_2$(FPM) values for all detectable genes from the IN VITRO BRADYZOITE gene set. (**D**) Mean centered, hierarchically clustered heatmap of log$_2$(FPM) values of the 24 genes that significantly differed in transcript abundance (fold-change ≥1; P$_{adj}$ <0.01) between control and pH 8.2 treated *H. hammondi*. As shown only three transcripts were found to be of higher abundance in pH-treated *H. hammondi*, while the remaining genes were of significantly lower abundance after pH exposure.

DOI: https://doi.org/10.7554/eLife.36491.017

The following figure supplement is available for figure 8:

*Figure 8 continued on next page*

*Figure 8 continued*

**Figure supplement 1.** Transcriptional profiling and qPCR of pH vs control RNAseq data.

DOI: https://doi.org/10.7554/eLife.36491.018

transcript abundance for *BAG1* (2 Days p=0.0261 and 3 Days p=0.0276), while *ENO1* and *LDH2* transcript levels remained statistically unchanged (*Figure 8—figure supplement 1B*). Taken together, our DBA staining, RNAseq and qPCR data strongly support the hypothesis that *H. hammondi* is refractory to alkaline pH induced stress at the transcriptional level, and this is correlates well with the complete lack of high pH-induced cyst wall formation in this species.

## Discussion

Parasites with multi-host life cycles undergo stereotyped patterns of development and growth in a given host, reaching a unique and often terminal developmental state that is then transmissible to the next host. With few exceptions, parasites with heteroxenous life cycles are obligately so; life stages that are infectious to one host (e.g., the definitive host) are not infectious to another host (e. g., an intermediate host). Among the Sarcocystidae (including such parasites as *Sarcocystis spp.*, *Besnoitia spp.*, *Neospora spp.*), all have exogenous life cycles for which they are obligately heteroxenous with the exception of *T. gondii*. Only in *T. gondii* oocysts and tissue cysts (i.e., bradyzoites) can infect, and ultimately be transmitted by, both intermediate and definitive hosts.

Given that the molecular determinants of life cycle dynamics such as those listed above are poorly understood in most parasite species, we have used the *T. gondii*/*H. hammondi* system as a means to better understand the genetic underpinnings of complex life cycles in Apicomplexa. In the present study, we conducted comparisons between one strain of *T. gondii* (VEG strain) and 2 strains of *H. hammondi* (HhCatEth1 and HhCatAmer), quantifying differences in in vitro and in vivo infectivity, replication rate, and in stage conversion. We identified dramatic differences in parasite biology between these two species that likely contribute to their significant life cycle differences.

### Stress-induced cyst formation is a unique trait in *T. gondii* compared to *H. hammondi*

While spontaneous cyst formation during growth in vitro has been reported for a handful of *T. gondii* strains (e.g., [*Jerome et al., 1998*; *Paredes-Santos et al., 2013*]), *T. gondii* of most genetic backgrounds can be induced to form tissue cysts in vitro through the application of a variety of chemical modifications (e.g., alkaline pH, nitric oxide donors, serum starvation; [*Jerome et al., 1998*; *Buchholz et al., 2013*; *Eaton et al., 2006*]) and these cysts are infectious to both the definitive and intermediate host (*Dubey, 1998*). *H. hammondi* was resistant to high pH exposure at the transcriptome level, and 100% resistant with respect to tissue cyst formation. This highlights a key distinction between *T. gondii* and *H. hammondi* biology: the dynamic nature of the *T. gondii* tachyzoite permits movement in both directions along the tachyzoite-bradyzoite continuum which is at least partly driven by sensing changes in the external environment. The ability to dynamically alter (and even delay) stage conversion may be made possible by the existence of *T. gondii* effectors that block responses to host immune effectors like IFN-γ (e.g., TgIST; [*Olias and Sibley, 2016*]) or otherwise manipulate the host response. If the progression from replicating *H. hammondi* zoite to tissue cyst is as predictable in vivo as it is in vitro (which is unknown but likely given that even IFN-γ knockout mice survive infection with *H. hammondi*; [*Dubey and Sreekumar, 2003*]), and *H. hammondi* has non- or less-functional orthologs of genes like *TgIST* (*Hakimi et al., 2017*; *Olias et al., 2016*; *Gay et al., 2016*), this may prevent the fixation of any polymorphisms in the population that might result in the suppression of spontaneous bradyzoite conversion. Given that bradyzoites induce less inflammation (*Kim et al., 2007*) and are protected from immune assaults by the tissue cyst wall (*Tu et al., 2017*; *Alday and Doggett, 2017*), the timing of these events may be under very strong selective pressure in *H. hammondi*. It will be interesting to see how the developmental dynamics of *H. hammondi* in vivo are timed with that of the intermediate host immune response, particularly in rodents (*Sheffield et al., 1976*; *Frenkel and Dubey, 1975*; *Mason, 1978*), since this may have directly impacted the timing of cyst formation during the evolution of this parasite. It is likely that *H. hammondi* is capable of sensing the stresses that are induced upon high pH exposure of the host

cell (as evidenced by significantly reduced transcript abundance for 24 *H. hammondi* genes), but if so it is clear that this species responds quite differently than *T. gondii*.

The link between stage conversion and parasite replication (*Radke et al., 2003*) makes it possible that the lack of cyst wall formation after pH stress in *H. hammondi* is because the parasite is arrested in a phase of the cell cycle that prevents conversion. However, in the present study there were no significant differences in vacuole and/or tissue cyst size between *T. gondii* and *H. hammondi* after pH exposure, indicating that parasite replication continued in both species at a similar (albeit slightly slower) rate. Moreover in *T. gondii* stress-induced (*Bohne et al., 1994*) or spontaneous (*Jerome et al., 1998*) cyst formation, slowed replication is a requirement for expression of brady-zoite antigens and ultimately stage conversion.

## *H. hammondi* possesses unique life stages

When taken together with the dynamics of spontaneous differentiation in *H. hammondi*, these data further demonstrate that *H. hammondi* terminally differentiates into a unique life stage that is no longer infectious to anything but the definitive host. While we did not conduct any formal analyses of the cell cycle stage(s) found within replicating and tissue cyst-like forms of *H. hammondi*, we would predict that spontaneous *H. hammondi in vitro*-derived tissue cysts form after a set number of division and would be homogenous in terms of cell cycle stage (likely $G_0$ and 1N) given their terminal differentiation phenotype. It is also not known if terminally differentiated parasites are capable of reinvading neighboring cells after egress, nor if they could be coaxed to infect cat gut epithelial cells in vitro, although the existence of cat intestinal organoids (e.g., [*Powell and Behnke, 2017*]) might allow for this to be tested directly in an accessible in vitro system.

Some insight into the *H. hammondi* developmental processes underlying these unique life stages can be gleaned from the D4 and D15 transcriptional profiling data. *H. hammondi* consistently had higher transcript levels for genes previously defined as being uniquely expressed in *T. gondii* brady-zoites (whether in vivo or in vitro) as early as 4 DPI, indicating that there is a high degree of similarity between the spontaneous differentiation process that occurs in *H. hammondi* and the stress-induced process observed in *T. gondii* (*Jerome et al., 1998*; *Croken et al., 2014a*; *Croken et al., 2014b*; *Weiss and Kim, 2000*; *Lyons et al., 2002*; *Behnke et al., 2008*). Importantly this enrichment for bra-dyzoite genes became more robust by 15 DPI compared to 4 DPI, indicating that the parasites are changing their transcriptional profiles throughout in vitro development (rather than becoming less viable or dying). Less expected, however, was the higher expression of genes in *H. hammondi* (observed in 24 hr PI, 4 DPI, and 15 DPI zoites) that have been found previously to be expressed more abundantly in *T. gondii* cat enteroepithelial stages and sporozoites (*Behnke et al., 2014*; *Hehl et al., 2015*). While the exact role of these gene products in determining life cycle and growth differences between *T. gondii* and *H. hammondi* are unknown, this provides strong evidence that sporozoite-derived life stages of *T. gondii* and *H. hammondi* are distinct, at least when grown in vitro. Importantly this enrichment becomes more apparent at 15 DPI, indicating that this is not entirely due to residual transcript from the sporozoite stage. Assuming that the majority of these transcript expression differences are manifested at the protein level (which is expected based on multiple studies in *T. gondii*; [*Wang et al., 2017*; *Xia et al., 2008*]), it is exciting to consider what impact these distinct expression profiles might have on parasite development and even the host response. Expression of these *T. gondii* merozoite-specific genes in *H. hammondi* zoites may reflect its terminally differentiated state, being infectious only the felid host (*Dubey and Sreekumar, 2003*). At a minimum our data may implicate only a subset of so-called 'merozoite-specific' genes (*Behnke et al., 2014*; *Hehl et al., 2015*) as being truly key distinguishing features of the *T. gondii* merozoite, since a number of them are highly expressed in *H. hammondi* zoites growing in HFFs.

## Developmental predictability allows for genetic manipulation in *H. hammondi*

In contrast to previously published work, we identified a short but very predictable replicative win-dow during which *H. hammondi* could be used to infect new host cells and identified the precise tim-ing of what appears to be an inevitable developmental program towards cyst formation. This finding is in contrast to previous studies where subculture was unsuccessful (*Sheffield et al., 1976*; *Riahi et al., 1995*), although in these studies subculture was attempted after 1, 2, 4, and 6 weeks of

growth and therefore most were outside of the window of infectivity we identified for *H. hammondi*. We exploited this window to genetically manipulate *H. hammondi* using transfection and gene editing strategies developed for *T. gondii* (*Shen et al., 2014*). Our transgenic *H. hammondi* parasite line expresses dsRED (30–40% after the first round of selection) and is >99% resistant to FUDR. This represents a new tool in the arsenal to track *H. hammondi* biology in vitro and in vivo, and opens up the door for the development of genetic screens to potentially disrupt spontaneous differentiation in *H. hammondi* to generate a mutant that can be cultured indefinitely, as well as utilized in gene-by-gene phenotypic screens.

## Replication rate and tissue cyst formation may contribute to the limited virulence and dissemination of *H. hammondi*

The nearly 3-fold difference in replication rate is likely a major determinant of reduced *H. hammondi* virulence (*Table 1*, *Figure 2D,E*), assuming this phenotype is recapitulated in vivo. Besides the impact that a slower replication rate would have on acute virulence, it may also impact infection outcome in the chronic phase. While *H. hammondi* parasites can at times be detected in the brains of mice after infection with oocysts (*Dubey and Sreekumar, 2003*), central nervous system (CNS) infection which is a hallmark of *T. gondii* infection (*Dubey and Sreekumar, 2003*; *Shen et al., 2014*) is much less common after exposure to *H. hammondi* (although *H. hammondi* parasites have been detected in mouse brains [*Dubey and Sreekumar, 2003*]). A simple explanation for reduced CNS infection in *H. hammondi* is that it never reaches a high enough burden to infect the CNS at any significant level. It is also possible that *H. hammondi* has additional deficiencies that underlie differences in tropism (such as poor trafficking by innate immune cells and/or inability to cross the blood-brain barrier; [*Konradt et al., 2016*; *Lambert et al., 2006*; *Sanecka and Frickel, 2012*]), although these have yet to be thoroughly explored. In contrast to differences in replication rate, we observed no substantial differences in the ability of *T. gondii* and *H. hammondi* to invade and form vacuoles within HFFs, suggesting that their invasion machinery is mostly conserved and intact. This idea finds some support in our transcriptome data, where we detected no enrichment for gene sets consisting of microneme, rhoptry or dense granule genes, and detected similar amounts of crucial invasion genes like AMA1, SpAMA1, as well as multiple rhoptry neck proteins (see *Supplementary file 1* for normalized RNAseq data). Moreover all genes known to be a part of the cellular invasion machinery appear to be well-conserved in both sequence and expression between these species ([*Lorenzi et al., 2016*]; *Supplementary file 1*).

## The potential for human exposure to and infection by *H. hammondi*

*T. gondii* and *H. hammondi* oocysts co-occur in the natural environment. A study conducted in Germany and surrounding countries found that the number of feline fecal samples that contained either *T. gondii* or *H. hammondi* oocysts was strikingly similar (*Schares et al., 2016*), meaning that humans would be just as likely to encounter *H. hammondi* oocysts as those of *T. gondii*. While more studies are needed to determine if co-occurrence is a widespread phenomenon, it raises important questions about the likelihood that *H. hammondi* is capable of infecting humans. Importantly, at present all serological tests would be expected to misdiagnose a *H. hammondi* infection as a *T. gondii* infection due to antibody cross-reactivity (*Dubey and Sreekumar, 2003*). While this is a very poorly explored line of research based on our data we would predict that an infection with *H. hammondi* should be less lethal than one with *T. gondii* since we would not expect a significant amount of recrudescence to occur. To further address the question of human infection and associated disease, mouse reactivation models and the development of differential diagnostic serological tests will be required in order to further determine human and animal infection rate differences between *H. hammondi* and *T. gondii* in the field.

## Materials and methods

### Parasite strains and oocyst isolation

Oocysts of *Hammondia hammondi* strain HhCatEth1 (*Dubey et al., 2013*), HhCatAmer (*Frenkel and Dubey, 1975*) and *Toxoplasma gondii* strain VEG (*Dubey, 2001*) were isolated from experimentally infected cats as described previously (*Dubey, 2001*). Briefly, wild type (for *T. gondii*) or interferon-γ

(IFN-γ) knockout (for *H. hammondi*) mice were orally infected with $10^4$ sporulated oocysts and sacrificed 4–6 weeks post-infection, and leg muscle (for *H. hammondi*) or brain (*T. gondii*) tissue were fed to 10–20 week old specific pathogen-free cats. Feces were collected during days 7–11 post-infection, and unsporulated oocysts were isolated via sucrose flotation. Oocysts were placed at 4°C in 2% $H_2SO_4$ to allow for sporulation to occur and for long-term storage.

Due to the limited availability and difficulty associated with the generation of oocysts, two strains (HhCatEth1 and HhCatAmer) of *H. hammondi* generated from multiple cat-derived oocysts preparations were used in experiments described below. Furthermore, various MOIs were utilized throughout the experiments described below to compensate for differences in replication between *T. gondii* and *H. hammondi*.

## Excystation of *T. gondii* and *H. hammondi* oocysts

Sporulated oocysts were washed 3X in Hanks' balanced salt solution (HBSS; Life Technologies, 14175145) and treated with 10% bleach in PBS for 30 min while shaking at room temperature. Washed pellets were resuspended in 3 mL of HBSS in a 15 mL falcon tube, and 4 g sterile glass beads (710–1,180 µM; Sigma-Aldrich, G1152) were added. Parasites were vortexed on high speed for 15 s on/15 s off, 4X. Supernatant was removed and pelleted by centrifugation. The pellet was resuspended and syringe-lysed using a 25 gauge needle in 5 mLs of pre-warmed (37°C) and freshly made, sterile-filtered excystation buffer (40 mL PBS, 0.1 g Porcine Trypsin (Sigma-Aldrich, T4799), 2 g Taurocholic Acid (Sigma-Aldrich, T4009), pH 7.5). After 45 min in a 37°C water bath, the suspension was syringe lysed again, and 7 mLs of cDMEM (100 U/mL penicillin, 100 µg/mL streptomycin, 2 mM L-glutamine, 10% FBS, 3.7 g $NaH_2CO_3$/L, pH 7.2) was added to quench the excystation media. The mixture was centrifuged, the pellet was resuspended in cDMEM, and used in downstream applications described in more detail below.

## Fixing parasites for immunofluorescence assays

Parasites were washed twice with PBS, fixed with 4% paraformaldehyde in PBS (Affymetrix, 19943), washed twice with PBS, and blocked/permeabilized in blocking buffer (50 mL PBS, 5% BSA, 0.1% Triton X-100).

## Quantification of sporozoite viability and replication rate

After 10 min incubation on HFFs at 37°C in 5% $CO_2$, monolayers containing freshly excysted parasites were scraped, serially syringe lysed (25 and 27 gauge needles), and pelleted. The pelleted parasites were resuspended in cDMEM, filtered through a 5 µM syringe-driven filter (Fisher Scientific, SLSV025LS) and used to infect 4-chambered slides (Lab-Tek II, 154526) containing a confluent monolayer of HFFs at an MOI of 0.5 for both *H. hammondi* CatEth1 and *T. gondii* VEG. After 1, 2, and 3 days of growth, cells and parasites were fixed as described above and stored in blocking buffer at 4°C until needed. Cells were immunostained with goat polyclonal *Toxoplasma gondii* Antibody (ThermoFisher Scientific, PA1-7256, RRID: AB_561769) at 1:500. The secondary antibody, Alexa Fluor 594 Donkey anti-Goat IgG (H + L) (ThermoFisher Scientific, A-11037, RRID: AB_2534105) was used at 1:1000. Coverslips were mounted to 4-chambered slides using VECTASHIELD Antifade Mounting Medium with or without DAPI, depending on the application (Vector labs, H-1000).

Vacuole formation was used as a proxy for sporozoite invasive capacity. To quantify this, the number of vacuoles with at least one parasite was determined in 100 random FOVs (1000x magnification) for three technical replicates for each parasite species at 1, 2, and 3 DPI. The entire experiment was repeated twice with different cat-derived oocyst preparations. For replication rate quantification, images of parasite-containing vacuoles were taken with an Axiovert 100 inverted fluorescent microscope with Zen lite 2012 software. The number of parasites per vacuole was determined for each image (biological replicates 1 and 2), and vacuole size (Biological replicate one only) was determined for each image using ImageJ software (NIH).

## Quantification of spontaneous *Dolichos biflorus*-positive cyst formation in vitro

For *Dolichos Biflorus* Agglutinin (DBA) staining, monolayers containing freshly excysted parasites were scraped, syringe lysed, and pelleted 24 hr after growth. The pellet was resuspended in

cDMEM, filtered through a 5 μM syringe-driven filter (Fisher Scientific, SLSV025LS), and passed at MOIs of 0.3 (HhCatEth1) and 0.001 for (TgVEG) onto coverslips containing a confluent monolayer of HFFs. Three coverslips were infected per strain for each time point analyzed. At 4 and 15 DPI, cells and parasites were fixed as described above and stored in blocking buffer at 4°C until immunostaining was conducted.

In addition to being stained with rabbit *Toxoplasma gondii* Polyclonal Antibody (Invitrogen, PA1-7252 RRID: AB_561769) at a dilution of 1:500 and a 1:1000 dilution of Alexa Fluor 594 goat anti-rabbit IgG (H + L) (ThermoFisher Scientific, A-11037, RRID: AB_2534095), coverslips were stained with Fluorescein labeled *Dolichos biflorus* Agglutinin (DBA; Vector Labs, FL-1031) at a dilution of 1:250. Coverslips were then mounted to microscope slides using ProLong Diamond Antifade Mountant with DAPI (ThermoFisher Scientific, P36962).

The number of DBA positive vacuoles and total vacuoles were quantified in 20 FOVs in three coverslips for each parasite at 4 and 15 DPI. Images were obtained with an Axiovert 100 inverted fluorescent microscope with Zen lite 2012 software and edited using ImageJ software (NIH).

For time course tissue cyst formation analysis, freshly excysted and filtered HhCatAmer and TgVEG sporozoites were used to infect confluent monolayers of HFFs grown on glass coverslips at an MOI of 0.5. After three days of growth, the media was changed every 2 to 3 days. At designated time points, coverslips were was twice with PBS, fixed with 4% formaldehyde in PBS, washed twice with PBS, and stored in blocking buffer 4°C until immunostaining was conducted using Rhodamine labeled DBA. The number of DBA positive vacuoles and number of total vacuoles were quantified in 15 parasite containing FOVs for each time point.

## Induction of bradyzoite formation in *T. gondii* and *H. hammondi*

Monolayers containing freshly excysted parasites were scraped, syringe lysed, and filtered through a 5 μM syringe-driven filter (Fisher Scientific, SLSV025LS) after 24 hr of growth. Filtered parasites were pelleted and passed at MOIs of 0.5 (*H. hammondi*) and 0.1 for (*T. gondii*) onto coverslips containing a monolayer of HFFs. Three coverslips were infected per parasite per treatment group. After 2 days, media was changed to pH 8.2 bradyzoite switch media (DMEM with 100 U/mL penicillin, 100 μg/mL streptomycin, 2 mM L-glutamine 10 mM HEPES, 2 g/L NaHCO$_3$, and 1% FBS [*Gay et al., 2016*]) or cDMEM. Coverslips with pH 8.2 media were grown at 37°C in the absence of CO$_2$. Media was changed again at 3 DPI. After 4 DPI, cells and parasites were fixed as described above and stored in blocking buffer at 4°C until needed. DBA and counter-staining was conducted using Fluorescein or Rhodamine labeled DBA and rabbit *Toxoplasma gondii* Polyclonal Antibody as described above. The number of DBA-positive vacuoles was quantified in 15 FOVs in three coverslips for each strain grown at either pH 7.2 or pH 8.2 and the percentage of DBA positive vacuoles was determined. Images were obtained and analyzed as described above.

## Subculture of *H. hammondi*

Sporozoites were obtained using the excystation protocol described above with the exception of syringe lysis and previously described host cell incubation. Six confluent monolayers of HFFs grown in T-25's were infected with *H. hammondi* American (HhCatAmer) sporozoites at an MOI of ~2 (2.8 million sporozoites). After a three-hour incubation at 37°C, the Day 0 Transfer flask was scraped, syringed lysed 5X with 25 gauge needle, filtered through a 5 μM syringe-driven filter (Fisher Scientific, SLSV025LS), and passed to a new confluent monolayer of HFFs grown in a T-25. This process was repeated after 2, 5, 8, 11, and 15 days post-excystation. After 3 days of growth following passage, 4.17 cm$^2$ of the T-25 was monitored daily. Images of vacuoles were obtained with an Axiovert 100 inverted fluorescent microscope with Zen lite 2012 software with a 40X objective, and vacuole sizes were quantified using ImageJ software (NIH). Similar methods were used to generate parasite samples for RNAseq analysis, except cells were processed for RNA harvest using methods described below.

## Characterizing limits of in vivo infectivity of *H. hammondi* grown in vitro

Sporozoites were obtained using the excystation protocol described above. After 24 hr, HFF monolayers infected with excysted sporozoites were scraped, syringe lysed 5X with a 25 gauge needle,

and pelleted. The pellet was resuspended in cDMEM, filtered through a 5 µM syringe-driven filter (Fisher Scientific, SLSV025LS) and passed onto confluent monolayers of HFFs grown in T-25s. Infected host cells were incubated at 37°C 5% $CO_2$ from 6, 8, 10, 13, and 15 days. The media was replenished after 5–7 days. At each time point, infected host cells were scraped, syringe lysed 3X with a 25 gauge needle and 3X with a 27 gauge needle, and pelleted. Parasites were counted and diluted in PBS, and a dose of 50,000 parasites was injected intraperitoneally in 2 BALB/C mice for each time point. Serum samples were obtained from infected mice daily for 9 days post-infection and the mass of the mice was also monitored daily. After 9 days of infection, the mice were sacrificed and dissected. Tissue samples were preserved in 10% neutral buffered formalin (Sigma HT501128) until immunohistochemistry analysis was performed.

## Tissue sectioning and staining

Fixed tissue samples were embedded in paraffin, sectioned, and stained using rabbit anti-*Toxoplasma* antibody (ThermoScientific Cat# RB-9423-R7) by Research Histology Services at the University of Pittsburgh. For antigen retrieval, deparaffinized slides were steamed at pH = 6.0 in 10 mM Citrate buffer. After exposure to 3% $H_2O_2$, slides were washed in Tris-buffered saline with 2.5% Tween-20 (TBST), and blocked for 20 min each with Avidin and Biotin blocking reagents (Vector labs; SP-2001) with TBST washes in between. Slides were blocked in 0.25% casein:PBS for 15 min, and incubated overnight in primary antibody (1:100 dilution in 3% goat serum in PBS). Slides were washed in PBST and incubated for 30 min with biotinylated goat anti-rabbit (Vector laboratories; BA-1000; 1:200 dilution in 3% goat serum in PBS). Following three washes with PBST, slides were incubated for 30 min with streptavidin-HRP (Vector laboratories; PK-6100), washed 3X with PBST, and incubated with AEC substrate (Skytec; ACE-500/ACD-015) for 15 min. Following rinses in water, slides were counterstained with aqueous hematoxylin and blued using Scott's tapwater substitute. Slides were mounted in Crystal Mount.

## Interferon-gamma (IFNγ) ELISA

Blood samples were obtained from mice daily via submandibular bleed, allowed to clot for 4–24 hr at 4°C, and centrifuged at 100 x g. Serum was stored at −20°C until ELISAs were performed. IFN-γ levels were determined using the BD OptEIA Set Mouse IFN-γ kit (Cat.# 555138) according to manufacturer's instructions. Serum samples were diluted 1:20.

## Transfection of *Hammondia* parasites and selection of recombinant parasites

Excysted sporozoites were prepared as described above and incubated overnight in a T-25 flask with confluent monolayer of HFFs. After 24 hr, the monolayer was scraped, syringe lysed 3X with a 25 gauge and 27 gauge needle, and filtered using a 5 µM syringe-driven filter (ThermoFisher Scientific, Pittsburgh, Pennsylvania, USA; SLSV025LS). The filtered contents were pelleted by centrifugation at 800 x g for 10 min, and the pellet was resuspended in 450 µl of cytomix with 2 mM ATP and 5 mM glutathione. Resuspended parasites were transferred to a cuvette, electroporated at 1.6 KV and a capacitance of 25 µF, and used to infect confluent HFF monolayers on coverslips. The coverslips were fixed 5 DPI as described above and mounted using ProLong Diamond Antifade Mountant with DAPI (ThermoFisher Scientific, P36962).

To create stable transgenic *H. hammondi* parasite lines, the CRISPR/CAS9 plasmid expressing a gRNA sequence targeting the uracil phosphoribosyl transferase gene (*UPRT*; [*Shen et al., 2014*]; kindly provided by David Sibley, Washington University) and a repair template consisting of 20 bp *UPRT*-targeting sequences flanking a PCR-amplified a dsRED expression cassette driven by the *T. gondii GRA1* promoter with a *GRA2* 3' UTR (amplified from the dsRED:LUC:BLEO plasmid described in (*Jeffers et al., 2017*; *Liao et al., 2013*) were used for transfection. Following centrifugation (800 x g, 10 min), 24 h *H. hammondi* zoites were resuspended in 450 µl of cytomix with ATP and glutathione containing 20 µg of the CRISPR/CAS9:*UPRT*:gRNA plasmid and 20 µg of PCR2.1 TOPO vector containing the dsRED repair template. The parasites were electroporated as above and each transfection contained at least 4 million parasites. Transfected parasites were then transferred to HFFs and grown for 2 days in cDMEM, then selected for 3 days by incubation in cDMEM containing 10 µM FUDR (*Figure 5B*). Parasites were again scraped, syringe lysed and filtered, and dsRED-

expressing parasites were collected in PBS using flow cytometry. Sorted parasites were injected intraperitoneally into 2 BALB/c mice. After 3 weeks of infection, mice were euthanized, skinned, and the intestines were removed before feeding to specific pathogen-free cats. The oocysts were collected and purified as described above, and oocysts, sporozoites and replicating parasites were evaluated for dsRED fluorescence using microscopy, flow cytometry and FUDR resistance.

## FUDR resistance of transgenic parasites

To test for deletion of the *UPRT* gene we quantified transgenic *H. hammondi* resistance to 5-fluoro-2'-deoxyuridine (FUDR). To do this, sporozoites of wild type and dsRED-expressing parasites were incubated overnight in a T25 flask with confluent monolayer of HFFs. After 24 hr, parasites were isolated by needle passage and filtration as above. We infected coverslips containing confluent HFFs with 50,000 parasites, and parasites were exposed to media alone or media containing or 20 µM FUDR. The parasites were allowed to grow for 4 days and then fixed using 4%PFA for 20 min. After blocking overnight in PBS/BSA/triton, coverslips were stained with rabbit anti-*Toxoplasma* antibody at a 1:500 dilution for an hour, and stained with Alexa-fluor 488-labeled goat anti Rabbit antibody (RRID: AB_143165). For each treatment/strain combination, we counted the number of parasites in at least 100 vacuoles, and for HhdsRED parasites we also quantified growth in both red and wild type vacuoles (since the HhdsRED parasites were from a mixed population).

## Genomic DNA isolation

To isolate genomic DNA from *H. hammondi*, freshly excysted parasites were used to infect HFFs. Following 7 days of growth, infected host cells were scraped, syringe lysed, pelleted and resuspended in 200 µL of PBS. Genomic DNA was isolated with the GeneJET genomic DNA Purification kit (ThermoFisher, K0721) according to the manufacture. Alternatively, identical procedures were used on freshly excysted and filtered sporozoites.

## Linear amplification of HhamdsRED gDNA

We used the Genomiphi Linear DNA amplification kit to linearly amplify DNA taken from our population of dsRED-expressing H. hammondi American parasites. We used ~5 ng of input DNA in a single reaction, and ethanol precipitated amplified DNA prior to quantification and use in PCR reactions.

## PCR verification of dsRED cassette

To verify that the dsRED cassette incorporated into the HhCatAmer genome, linearly amplified pure parasite genomic DNA (HhamdsRED) was amplified using primer sets targeting the UPRT homology arms of the dsRED cassette, the dsRED transgene, and a sequence specifically found in *H. hammondi* (*Sreekumar et al., 2005*). (See *supplementary file 3* for primer sequences) Controls included wild type HhCatAmer genomic DNA, purified from infected HFFs, and a water control. PCR reactions were conducted using 12.5 µL of 2X BioMix Red (Bioline, BIO-25006), 1 nM of both forward and reverse primers, 45 ng of template DNA, and water to 25 µL. Cycling conditions began with 5 min of 95°C denaturation, followed by 30 cycles of 95°C for 30 s, 50°C for 30 s, 72°C for 1 min and 30 s, followed by 72°C for 5 min and a 4°C hold. PCR products were visualized on a 1% agarose gel.

## RNAseq data collection and enrichment analysis for D4 and D15 zoites

Sporozoites were isolated from HhCatEth1 and TgVEG sporulated oocysts as described above and used to directly infect confluent HFF monolayers in 96 well tissue culture plates with MOIs of 3, 1 or 0.5 purified sporozoites for each species. On day four post-infection, wells were observed and chosen for RNA analysis based on similar numbers of parasite-containing vacuoles lack of significant host cell lysis in both species. For the Day four samples, the MOI = 3 infected wells were chosen for *H. hammondi*, while the MOI = 0.5 infected wells were chosen for *T. gondii*. Wells were washed 3X with ~200 µL cDMEM, and RNA was harvested using Trizol (Invitrogen). Remaining *H. hammondi* wells were washed with 3X with cDMEM on Day four and then again on Day nine prior to harvest on Day 15 post-infection. For *T. gondii*, remaining MOI = 3 and 1-infected wells were scraped and syringe lysed on Day four and Day nine and used to infect new monolayers in 96 well plates at an MOI of either 0.5 (Day 4) or 0.3 (Day 9). On Day 15, wells for both *H. hammondi* (not subcultured) and *T. gondii* (subcultured 2x) were washed 1X with cDMEM and then harvested for RNAseq

analysis. Two samples were harvested for each species on Day 4, and three samples were harvested from each species on Day 15. Total RNA samples were processed for Illumina next generation sequencing using the Mobioo strand-specific RNAseq library construction kit. Samples were analyzed on an Illumina Nextseq and demultiplexed using NextSeq System Suite software. Reads were aligned to their respective species genome assembly (*H. hammondi* v10 or *T. gondii* strain ME49 v10; toxodb.org; [*Lorenzi et al., 2016*]) using the Subread package for Linux (v. 1.4.6; [*Alday and Doggett, 2017*]) with subread-align using default parameters except for –u to keep only uniquely mapping reads. featureCounts from the subread package (*Alday and Doggett, 2017*) was used to quantify the number of mapping reads per transcript, using default settings except for –s 2 (for stranded read mapping), –t CDS, –g Parent (for specific compatibility with the *T. gondii* and *H. hammondi* gff files), –Q 10 (minimum mapping quality required). The –t CDS option was chosen for both *T. gondii* and *H. hammondi* because to date no 5′ or 3′ UTR sequences have been predicted for *H. hammondi* (in contrast to *T. gondii*). Fastq files have been deposited in the NCBI short read archive (SRX3734444-SRX3734449).

## RNA sequencing and differential expression analysis of 24h-infected THP-1 cells (a human monocyte cell line)

To validate genesets that were significantly enriched in D4 and D15 *H. hammondi* compared to *T. gondii*, we extracted sporozoites from oocysts and grew them overnight in HFFs as described above. Parasites were released by scraping and needle passage as above, and then passed through a 5 µm syringe-driven filter (Fisher Scientific, SLSV025LS). Filtered parasites were used to infect monocytes (THP-1, RRID: CVCL_X588) at MOI of 4. At 24 hr post-infection, RNA was harvested using RNeasy Mini Kit (Qiagen; three samples for each species). Total RNA samples were processed for Illumina next generation strand-specific sequencing using the Illumina TruSeq stranded mRNA sequencing kit. Fastq files have been deposited in the NCBI short read archive (SRX3734421-SRX3734428). FASTQ reads were aligned to their respective species genome assembly (*H. hammondi* release 34 or *T. gondii* strain ME49 release 30; toxodb.org) with CLC Genomics Workbench v10.0.1 (http://www.qiagenbioinformatics.com/) using default parameters for reverse strand mapping. Raw count data (with at least one read in at all samples for HhCatAmer) generated by CLC Genomics Workbench were loaded into R statistical software and analyzed using the DESeq2 package (*Varet et al., 2016*). Read count data of HhCatAmer were compared to TgVEG to identify genes of different abundance and genes were deemed differentially expressed according to the threshold of $\log_2$ (fold) >1 and $P_{adj}$ <0.01. $\log_2$ transformed data were used for enrichment analysis using Gene Set Enrichment Analysis (GSEA) (*Subramanian et al., 2005*). Data were compared to previously curated gene sets (434 total; as published in [*Behnke et al., 2014*]) as well as seven additional gene sets that we curated ourselves (listed in *Table 2*). All enrichment profiles were deemed significant if the FDR q-value was ≤0.01. We also identified 1069 genes that were of different abundance between HhCatAmer and TgVEG at this 24 hr time point in THP-1 cells.

## RNAseq from pH-treated *T. gondii* and *H. hammondi* zoites

Sporozoites were isolated from HhCatAmer and TgVEG sporulated oocysts as described above and used to directly infect confluent HFF monolayers in 24 well tissue culture plates with MOI of 1.3 for HhCatAmer and an MOI of 1 for TgVEG. Infected host cells were maintained as described the induction of bradyzoite section above. On day four post-infection, wells were washed twice with 1X PBS and RNA was harvested using the RNeasy Kit according to the manufacturer (Qiagen, 74104) using QiaShredder spin columns (Qiagen, 79654) to homogenize samples, and RNase-free DNase to degrade contaminating DNA (Qiagen, 79254). Total RNA samples were ethanol precipitated and processed for Illumina next generation strand specific sequencing as described for infected THP-1 RNAseq experiments. Library preparation and read alignment were performed as described in infected THP-1 RNAseq experiments.

## Identification of differentially expressed genes using DESeq2 and gene set enrichment analysis

Raw count data per transcript (generated by featureCounts or CLC genomics) were loaded into R statistical software and analyzed using the DESeq2 package(*Love et al., 2014*). Comparisons of D4

and D15 read count data were used to identify transcripts of different abundance at each time point, and differences were deemed significant at $P_{adj}$ <0.05. Data were $log_2$ transformed and normalized using the *rlog* (with the blind = TRUE option) function in DESeq2 for use in downstream analyses. $Log_2$ transformed, normalized data from DESeq2 (hereafter referred to as $Log_2$ (FPM)) were analyzed for enrichment using Gene Set Enrichment Analysis (GSEA; [*Subramanian et al., 2005*]). Since read count overall from the *T. gondii* libraries were much greater than those from the *H. hammondi* libraries, we took the 7372 genes matched based on the previously published gene-by-gene annotation (*Lorenzi et al., 2016*) and selected only those that had at least 1 day 4 sample and 1 day 15 sample with >5 reads for the D4 and D15 data and samples that had reads for each replicate and a total read average of 5 reads per three replicates for pH 8.2 analysis. In total, 4146 genes for the D4 and D15 analysis and 907 genes for the pH 8.2 analysis passed these benchmarks and were used in subsequent analyses. We used this approach to identify gene sets that were significantly different between days 4 and day 15 in culture in both species and those that were different between species at both day 4 and day 15 and in pH 8.2 treated and control samples. We compared these data using previously curated gene sets (434 total; as published in [*Walzer et al., 2014*]) as well as seven additional gene sets that we curated ourselves. These are listed in *Table 2*. All enrichment profiles were deemed significant if the FDR q-value was ≤0.05.

## cDNA synthesis and qPCR

For qPCR analysis, after 24 hr of growth, monolayers containing excysted sporozoites were scraped, syringe lysed, and pelleted. The pellet was resuspended in cDMEM, filtered through a 5 µM syringe-driven filter (Fisher Scientific, SLSV025LS) and passed at MOIs of 1 (*T. gondii*) and 7 (*H. hammondi*) into 96-well plates or 24-well plates containing HFF monolayers. Mock-infected controls consisted of filtering parasites through a 0.22 µM syringe-driven filter (Fisher Scientific, SLGL0250S). Three replicates were made for each sample per isolation time point. RNA for D4 and D15 RNAseq validation was collected at day 4 and day 15 post-infection, (equivalent to day 5 and day 16 post-excystation) from parasites and mock infections grown in 96-well plates. RNA for pH induced bradyzoite transcript analysis was collected from day 4 and 5 post-infection samples grown in 24-well plates. RNA was collected using the RNeasy Kit according to the manufacturer (Qiagen, 74104) using QiaShredder spin columns (Qiagen, 79654) to homogenize samples, and RNase-free DNase to degrade contaminating DNA (Qiagen, 79254). Isolated RNA was ethanol precipitated, resuspended in RNase-free water, and these preparations were used to create cDNA using Superscript III Reverse Transcriptase Kit using Oligo(dT) primers according to the manufacturer (ThermoFisher Scientific, 18080051). All RNA samples were kept at −80°C, and all cDNA reactions were kept at −20°C. Prior to qPCR use, cDNA was diluted 1:10 (for RNAseq validation) or 1:5 (for bradyzoite transcript analysis) with $H_2O$.

qPCR assays were performed on a QuantStudio 3 Real-Time PCR System in a 10 µl reaction volume containing 5 µl 2x SYBR Green Master Mix (VWR International, 95030–216), 3 µl of cDNA template, 1 µl $H_2O$, and 1 µl 5 µM primer. Controls included a reverse transcription negative control and a water-template control. The thermal cycling protocol was 95°C, 10 min; 40 cycles of (95°C, 15 s; 60°C, 1 min); 4°C hold. The melt curve protocol was 95°C, 15 s; 60°C 1 min, 95°C 15 s. The control gene was dense granule 1 (GRA1), and samples were tested in duplicate or triplicate. Melt curves were performed on each plate (with the exception of 2 plates). Data were analyzed using the $2^{-\Delta\Delta Ct}$ method (*Livak and Schmittgen, 2001*), and statistical analyses were conducted on the $\Delta Ct$ values (as in [*Boyle et al., 2003*]).

## Animal statement

All mouse experiments were performed with 4- to 8-wk-old BALB/C and C57BL/6J mice. Animal procedures met the standards of the American Veterinary Association and were approved locally under Institutional Animal Care and Use Committee protocol no. 12010130.

## Data availability statement

RNAseq data has been deposited in the NCBI short read archive. Sequencing data generated and analyzed in this is available in *supplemental files 1* and *2*. Gene set categorization of significant

genes for all gene sets used in analyses in this study are available in **supplemental files 4** and **5**. All additional data is available in the manuscript and supporting files.

## Acknowledgements

The authors would like to thank Clara Stuligross for a critical reading of the manuscript. At the University of Pittsburgh, Cori Richards-Zawacki, Stephanie Ander, and Carolyn Coyne shared expertise, reagents, and equipment required for qPCR. This work was supported by R01AI116855 and R01AI114655 to JPB.

## Additional information

### Funding

| Funder | Grant reference number | Author |
| --- | --- | --- |
| National Institutes of Health | R01AI116855 | Jon P Boyle |
| National Institutes of Health | R01AI114655 | Jon P Boyle |

The funders had no role in study design, data collection and interpretation, or the decision to submit the work for publication.

### Author contributions

Sarah L Sokol, Conceptualization, Data curation, Formal analysis, Validation, Investigation, Methodology, Writing—original draft, Writing—review and editing; Abby S Primack, Conceptualization, Data curation, Formal analysis, Investigation, Methodology, Writing—original draft; Sethu C Nair, Data curation, Methodology; Zhee S Wong, Data curation, Software, Formal analysis, Methodology; Maiwase Tembo, Data curation, Investigation; Shiv K Verma, Resources, Methodology; Camila K Cerqueira-Cezar, Conceptualization, Resources, Methodology; JP Dubey, Jon P Boyle, Conceptualization, Resources, Data curation, Software, Formal analysis, Funding acquisition, Investigation, Methodology, Writing—original draft, Project administration, Writing—review and editing

### Author ORCIDs

Jon P Boyle http://orcid.org/0000-0003-0000-9243

### Ethics

Animal experimentation: All mouse experiments were performed with 4- to 8-wk-old BALB/C and C57BL/6J mice. Animal procedures met the standards of the American Veterinary Association and were approved locally under Institutional Animal Care and Use Committee protocol no. 12010130.

### Decision letter and Author response

Decision letter https://doi.org/10.7554/eLife.36491.031
Author response https://doi.org/10.7554/eLife.36491.032

## Additional files

### Supplementary files

• Supplementary file 1. D4D15 RNASeq and 24 hr THP-1 RNAseq. Xls file containing raw reads, rLog values, and results for DESeq2 analysis for D4 and D15 HFF infection with *H. hammondi* Eth1 and *T. gondii* VEG and 24 hr THP-1 infection with *H. hammondi* Amer and *T. gondii* VEG.
DOI: https://doi.org/10.7554/eLife.36491.019

• Supplementary file 2. pH vs Ctl RNAseq data. Xls file containing raw reads, rLog values, and results for DESeq two analysis for pH vs control treated *H. hammondi* Amer and *T. gondii* VEG HFF infection.
DOI: https://doi.org/10.7554/eLife.36491.020

- Supplementary file 3. Primer Table. File containing all of the primers used in this study, the purpose of the primer, and the Toxodb.org gene ID for each gene queried.
DOI: https://doi.org/10.7554/eLife.36491.021

- Supplementary file 4. Gene Set Categorization Significant Genes D4 vs D15 and 24 hr THP-1. Xls file containing all genes belonging to gene sets used in analyses and indicating which genes are significant in which categories for analysis. Significant genes for D4, D15, and 24 hr THP-1 data sets are indicated by the number 1, highlighted in a green box.
DOI: https://doi.org/10.7554/eLife.36491.022

- Supplementary file 5. Gene Set Categorization Significant Genes pH vs Ctl. Xls file containing all genes belonging to gene sets used in analyses and indicating which genes are significant in which categories for analysis. Significant genes for pH versus control data sets are indicated by the number 1, highlighted in a green box.
DOI: https://doi.org/10.7554/eLife.36491.023

- Transparent reporting form
DOI: https://doi.org/10.7554/eLife.36491.024

## Data availability

All raw RNAseq datafiles used in this study can be found at the NCBI short read archive (SRA) under project numbers PRJNA433890 and PRJNA433960. Sequencing data generated and analyzed in this is available in supplemental files 1 and 2. Gene set categorization of significant genes for all gene sets used in analyses in this study are available in supplemental files 4 and 5. All additional data is available in the manuscript and supporting files.

The following datasets were generated:

| Author(s) | Year | Dataset title | Dataset URL | Database, license, and accessibility information |
|---|---|---|---|---|
| Sarah L Sokol, Abby S Primack, Sethu C Nair, Zhee S Wong, Maiwase Tembo, Shiv K Verma, Camila K Cerqueira-Cezar, JP Dubey, Jon P Boyle | 2018 | Comparative transcriptomics of *Toxoplasma gondii* and *Hammondia hammondi* | https://www.ncbi.nlm.nih.gov/bioproject/?term=PRJNA433890 | Publicly available at NCBI BioProject (Accession No. PRJNA433890) |
| Sokol SL, Primack AS, Nair SC, Wong ZS, Tembo M, Verma SK, Cerquerira-Cezar CK, Dubey JP, Boyle JP | 2018 | Transcriptional profiling of *T. gondii* and *H. hammondi* during in vitro development | https://www.ncbi.nlm.nih.gov/bioproject/PRJNA433960 | Publicly available at NCBI BioProject (Accession no. PRJNA433960) |

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
