## [Decision Letter]

Thank you for submitting your work entitled "*Hammondia hammondi* has a developmental program in vitro that mirrors its stringent two host life cycle" for consideration by *eLife*. Your article has been reviewed by two peer reviewers, and the evaluation has been overseen by a Reviewing Editor and a Senior Editor. The following individuals involved in review of your submission have agreed to reveal their identity: Louis Weiss (Reviewer #2).

Our decision has been reached after consultation between the reviewers. Based on these discussions and the individual reviews below, we regret to inform you that your work will not be considered for publication in *eLife* at this point.

This study explores exciting features of *Hammondia hammondi* and holds great potentials for uncovering unexpected biology in this Apicomplexan compared to Toxoplasma. Despite being generally positive about the work, reviewer 1 raised serious issues in regard to the superficial and descriptive nature of some of the findings. The reviewers agreed that it will take some time (more than two months) to fill the gaps and provide the kind of conceptual and mechanistic advance that would warrant publication in *eLife*. While the paper is rejected as a result of this, we are broadly supportive of this work and enthusiastically encourage you to resubmit as a new manuscript if you can address the issues raised. In this case, we will consider it and treat as a revised manuscript. If you choose this course of action please submit a separate cover letter detailing the changes you have made.

*Reviewer #1:*

The manuscript "*Hammondia hammondi* has developmental program in vitro that mirrors its stringent two host life cycle" is both an exciting new chapter in the development of genetic models for apicomplexan parasites and a less exciting descriptive paper that largely confirms basic biological paradigms of *H. hammondi* (Hh). There are relatively few apicomplexans for which stable transgenic parasites can be produced due to limitations in the ability to grow these parasites in cell culture. Utilizing a combination of short term in vitro culture followed by infection of mice, Sokol et al., show convincingly they have generated *H. hammondia* transgenic parasites that are FuDR resistant and dsRED fluorescent. There are still issues that need to be addressed for these protocols to be fully useful including whether these transgenic parasites can be cloned, do other selection strategies from Toxoplasma (Tg) work, is recombination random or template-driven, and can a knockout or epitope-tagged gene be produced.

Other parts of in this paper either are largely confirming the conclusions of previous studies or are incomplete.

1) The first of these issues is why Hh cultures become exhausted. This is a well-known phenotype of Hh in vitro. Mechanisms of invasion, development that leads to growth arrest or just loss of viability could explain the exhaustion of these cultures, none of which were systematically dissected here. Thus, Figure 2 adds very little new to the field or this paper.

2) The second issue is whether Hh is resistant to methods of in vitro cyst induction. A single alkaline pH condition at one time point (48hr) was used to reach the broad conclusions about Hh induction-resistance. Yet, in Figure 3, these authors show that Hh takes four more days to spontaneously differentiate into cysts than TgVEG, which indicates testing these ideas with a short, single time point is insufficient. Even in Tg models, two days of alkaline-induction is often not enough time to see good tissue cyst numbers. There was no consideration of the toxicity of pH8.2 media and it was shown early on in the development of Tg bradyzoite models that blocking growth prevents tissue cyst differentiation. Thus, Figure 4 experiments are far from complete. To really pin this issue down, more time points in different pH media is needed, viability needs to be addressed, and alkaline-induction needs to be compared to other methods of induction.

3) Finally, the transcriptome experiments have significant problems. The small number of reads from in vitro cultured parasites mapping to the Hh genome requires large normalizations potentially introducing significant errors. The biological significance of the new finding that merozoite mRNAs are more expressed in Hh is not clear. The three highest expressed merozoite mRNAs are also significantly expressed in the Hh sporozoite (e.g. 253250, 277230, 287040), which is a product of the definitive cycle. It is not clear merozoite mRNA expression in these cultures represents a normal progression or slow mRNA turnover leading to carry over. The lack of evidence of protein expression also reduces the significance of this study. Thus, these results may have more to do with the limitations of parasite in vitro behavior than natural biology. Notwithstanding, the obvious technical issues, the main conclusion of these experiments, is that Hh expresses bradyzoite mRNAs earlier and at higher levels compared to Tg, which was published by this group in 2014. Given all the problems, this data needs to be consolidated and a much better discussion of the data and its caveats is also needed. Figure 8 and 9 could be combined into a single figure eliminating the outdated heat maps as they do not represent all the data and are redundant. Providing more organization and gene list breakouts along with more explanation of the columns and methods in Supplementary file is what is needed here.

*Reviewer #2:*

This is a very interesting paper that provides both an important new model to study Apicomplexan biology and provides the first evidence of genetic manipulation of *Harmmondia*. The paper is carefully done and well reasoned. The data are clearly presented. I have only three minor comments

1) It looks like replication goes for a set number of divisions before terminal differentiation. This is why transfer can be achieved but the program is still counting and the expansion post transfer correlates to the original start of the initial culture and does not reset to day 1 on transfer. This should be more clearly explained in the text

2) As to *T. gondii* Bradyzoites found in in vitro cysts exist in both Go and G1 stages. The G1 stages can replicate in cell cultures and resinfect the intermediate host and it is likely the G0 are limited to cat infection (similar to Harmmondia). This is supported by plaque assays from mouse brain cysts, only a subset of zoites that are counted in a heamocytometer can establish infection in vitro.

3) For the mouse experiment is there any data on expansion of the infection (e.g. is the final parasite load dependent on how long the culture has been in vitro) One would expect with a fixed division differentiation program this would be the case.

[Editors’ note: what now follows is the decision letter after the authors submitted for further consideration.]

The title needs to be changed to more accurately reflect the content of the manuscript and relevant finding. We suggest: "Dissection of the developmental program and foundational work to develop it as a new genetic model ". This change may be made in correspondence with the production staff as no further editorial oversight is required. Although reviewer #1 had remaining concerns, a discussion among the reviewers in a final consultation session resulted in this final decision to accept the work in its current form.

This report is an exciting new chapter in the development of genetic models for apicomplexan parasites and a comprehensive, experimental documentation of the basic biological paradigms of *H. hammondia*. There are relatively few apicomplexans for which stable transgenic parasites can be produced due to limitations in the ability to grow these parasites in cell culture. Utilizing a combination of short term in vitro culture followed by infection of mice, Sokol et al., provide the first evidence of genetic manipulation of Harmmondia by generating FuDR resistant and dsRED fluorescent parasites. *H hammondia* is not responsive to the stress-induced encystation which emerges as a unique trait in the closely related parasite Toxoxplasma gondii.

---

## [Author Response]

We appreciated the comments of both reviewers on the initial submission, and your willingness to allow us to resubmit the manuscript along with a point-by-point response to the original reviewer comment. We have made extensive changes and updates to the manuscript in line with the reviewer comments and we hope that you would again be willing to consider this manuscript for publication in *eLife*. Multi-host parasite life cycles are among the most intriguing evolutionary relationships in biology. *T. gondii* is unique among Apicomplexans in that asexual life stages are infectious to both the intermediate (all warm-blooded animals) and definitive (felines) hosts. The cellular and molecular mechanisms driving these differences are unknown.

To begin to identify and characterize these mechanisms we have looked to the nearest extant relative of *T. gondii, Hammondia hammondi*. Unlike *T. gondii, H. hammondi* is obligately heteroxenous, despite that these parasites share the same definitive host and many of the same intermediate hosts. We have thoroughly characterized the in vitro growth characteristics of *H. hammondi*, including its ability to be grown in tissue culture and in a mouse model of infection. This work resulted in novel insights into how *T. gondii* and *H. hammondi* may fundamentally differ at the cellular and molecular level, and identified sensitivity to stress as a key feature that distinguishes *T. gondii* and *H. hammondi*. […]

Overall, we have discovered key differences in the growth characteristics of *T. gondii* and *H. hammondi* that are likely relevant to their distinct life cycles. This work provides multiple lines of evidence that *H. hammondi* progresses through life stages that are completely distinct from those in *T. gondii*, despite having very similar gene content. This work has changed our thinking about stress responses and bradyzoite development in *T. gondii*, and how that ability can be completely decoupled from cyst formation. Overall this work provides new insights into T. gondii cyst biology, a life stage that is critical for both the transmission of the parasite and for its reactivation in immunosuppressed patients.

Reviewer #1:

The manuscript "Hammondia hammondi has developmental program in vitro that mirrors its stringent two host life cycle" is both an exciting new chapter in the development of genetic models for apicomplexan parasites and a less exciting descriptive paper that largely confirms basic biological paradigms of H. hammondi (Hh). There are relatively few apicomplexans for which stable transgenic parasites can be produced due to limitations in the ability to grow these parasites in cell culture. Utilizing a combination of short term in vitro culture followed by infection of mice, Sokol et al., show convincingly they have generated H. hammondia transgenic parasites that are FuDR resistant and dsRED fluorescent. There are still issues that need to be addressed for these protocols to be fully useful including whether these transgenic parasites can be cloned, do other selection strategies from Toxoplasma (Tg) work, is recombination random or template-driven, and can a knockout or epitope-tagged gene be produced.

The data we present in this manuscript show that we were able to generate a stable transgenic line of *H. hammondi* that has incorporated an exogenous DNA cassette into its genomic DNA even as our total number of replicative parasites decreases overtime. We see this as the most important innovation of our transgenic work with this parasite species, since the existing literature was not completely clear as to whether these types of manipulations would be possible (and in fact would suggest that it would not be possible).

We suspect that integration of the exogenous DNA is due to a random insertion of the cassette into the *H. hammondi* genome, since we could not detect evidence for recombination of the cassette at the CRISPR/CAS9 cut site (see Figure 6—figure supplement 1). We do show, however, that FUDR resistance is very tightly linked to dsRED fluorescence in our transgenic line, indicating that CRISPR/CAS9 expression along with the *UPRT*-targeting guide RNA was effective (and was nearly always associated with integration of the dsRED cassette). Therefore, our work shows that exogenously supplied plasmids can be inserted into the *H. hammondi* genome, and that multiple foreign genes can be expressed in this organism using *T. gondii* promoters (e.g., dsRED, CAS9 nuclease).

Based on our experience and the work outlined in this manuscript, direct cloning individual *H. hammondi* is likely not possible due to the inherent differences in the biology of this parasite compared to *T. gondii.* An individual *H. hammondi* parasite will not form an infectious plaque but rather a single terminally differentiated bradyzoite-containing cyst. While it is possible that a single cyst could successfully infect a cat and produce viable oocysts, we are not aware of any studies that have attempted this due to the likely high failure rate of single bradyzoite cyst infections. Therefore this approach would require more than the 1-2 cats required to generate a single *H. hammondi* transgenic line using the flow cytometry based approach described here, since the success rate of single bradyzoite infections in cats is known to be ~12.5 percent for the *T. gondii* VEG strain (1). We could pool individual cysts and use those to infect cats to increase the infection rate, but this approach would suffer from the same deficiencies as the flow cytometry approach, and would be even less efficient given the low numbers of parasites that could be isolated.

We do not yet know if homology-directed insertions are possible in this organism, and correspondingly whether we could perform endogenous tagging experiments. However, given the fact that both ectopic and homologous recombination activities are both extensively conserved in the closely related coccidian Apicomplexan *N. caninum* (e.g., see (2, 3)), we think it very unlikely that homology-based approaches would be ineffective in *H. hammondi* given its phylogenetic position. The experiments needed to evaluate exactly how transgenes are either incorporate or deleted from the parasite genome would require at least 1, if not 2, additional feline hosts per construct. While a thorough investigation of how genetic manipulation techniques used in *T. gondii* would help us better compare these species, such experiments would not contribute as much to the field as compared to the value of determining the impact of a specific gene editing/deletion on parasite biology. We would have trouble justifying these experiments from a purely mechanistic standpoint given the number of cats required for thorough downstream analyses of transgenic parasites.

Other parts of in this paper either are largely confirming the conclusions of previous studies or are incomplete.

1) The first of these issues is why Hh cultures become exhausted. This is a well-known phenotype of Hh in vitro. Mechanisms of invasion, development that leads to growth arrest or just loss of viability could explain the exhaustion of these cultures, none of which were systematically dissected here. Thus, Figure 2 adds very little new to the field or this paper.

Since in vitro derived tissue cysts (generated using nearly identical culture conditions as those described here) are orally infectious to the feline host (see Introduction section for citations), we do not think that loss of viability is the root cause for the changes that we see in vitro throughout development. Moreover invasion appears to be similar between parasite lines (at the sporozoite or 24 hour zoite stage), suggesting that different mechanisms of invasion are not the culprit. While the emergence of bradyzoite-like parasites during the cultivation of *H. hammondi* has been shown before, the current state of the literature has not characterized the precise timing of bradyzoite development in *H. hammondi*. Moreover, there are no reports in the literature comparing this developmental process to *T. gondii*, and specifically *T. gondii* VEG which is also known to spontaneously convert to tissue cyst stages with some frequency. It was entirely possible that the extent of bradyzoite conversion in *H. hammondi* was no different than *T. gondii* VEG. What we found instead was that the spontaneous conversion of *H. hammondi* and *T. gondii* VEG were distinct in both timing and degree. Moreover, no previous work has demonstrated that eventually the spontaneous conversion reaches 100% in *H. hammondi* (the few papers that have looked at this have not performed such a detailed analysis). We assert that this figure (which is now figure 1) in combination with Figure 2 are a very important contributions to our understanding of just how different *T. gondii* and *H. hammondi* are during in vitrodevelopment, and that 100% of all *H. hammondi* vacuoles become cysts during in vitro cultivation.

2) The second issue is whether Hh is resistant to methods of in vitro cyst induction. A single alkaline pH condition at one time point (48hr) was used to reach the broad conclusions about Hh induction-resistance. Yet, in Figure 3, these authors show that Hh takes four more days to spontaneously differentiate into cysts than TgVEG, which indicates testing these ideas with a short, single time point is insufficient. Even in Tg models, two days of alkaline-induction is often not enough time to see good tissue cyst numbers. There was no consideration of the toxicity of pH8.2 media and it was shown early on in the development of Tg bradyzoite models that blocking growth prevents tissue cyst differentiation. Thus, Figure 4 experiments are far from complete. To really pin this issue down, more time points in different pH media is needed, viability needs to be addressed, and alkaline-induction needs to be compared to other methods of induction.

To address this comment we have conducted additional time course experiments where infected host cells were treated with pH 8.2 media for 96 hours. Even in this extended treatment, we observe the same result as we observed with a 48 h exposure to high pH (not a single *H. hammondi* vacuole was found to be DBA-positive). Furthermore, under these conditions (as shown in Figure 7), we obtain as high as 80-90% tissue cyst formation in TgVEG. These results address the concerns of reviewer 1 that not enough time was allowed for tissue cyst formation, and are at the upper limit of time that host cells can tolerate being in high pH medium. Additionally, we addressed the issue that pH 8.2 treatment was preventing replication, a concern for inhibiting tissue cyst development, following 48 hours of pH 8.2 exposure by determining that there was no significant difference in the number of parasites per vacuole with species either exposed control of alkaline pH conditions. (See Figure 7B). Furthermore, we have now performed RNAseq and qPCR on pH exposed *T. gondii* and *H. hammondi* and in doing so find that the *H. hammondi* transcriptome is highly refractory to alkaline pH treatment (Figure 8). While we agree that investigating other methods of in vitro cyst induction would be interesting, pH is a well-established protocol to induce the formation of fully infectious tissue cyst stages in vitro for *T. gondii*. While there are some differences between tissue cysts formed in vivo and those generated in vitrousing this method, overall their gene expression profiles are quite similar. The fact that *H. hammondi* does not respond to alkaline pH at the transcriptional level is a novel phenomenon that represents a previously unknown phenotypic difference between these parasite species, and again indicates that the *H. hammondi* replicative stages are a distinct from *T. gondii* tachyzoites. The fact that very few transcripts change in abundance in *H. hammondi* during pH exposure demonstrates quite dramatically that there is a key difference between *T. gondii* and *H. hammondi* at the level of stress signal detection and/or processing. This signal detection and processing network is unknown in *T. gondii*, and our manuscript establishes the *T. gondii*/*H. hammondi* comparative system as a novel means of identifying this network.

3) Finally, the transcriptome experiments have significant problems. The small number of reads from in vitro cultured parasites mapping to the Hh genome requires large normalizations potentially introducing significant errors.

We agree that the small number of reads from in vitro parasites mapping to both the *H. hammondi* and *T. gondii* genome are of concern, and addressed this head on in the first submission as well as in this revision. To further address this issue we have performed additional qPCR and RNAseq analyses in order to verify our differential expression results derived from our RNAseq experiments (See Figure 3—figure supplement 3.) We certainly appreciate this concern but feel confident in the reproducibility of our results using both RNAseq and qPCR. Moreover, as we discuss immediately below, we have performed follow up RNAseq assays at the 24 hour time point and see very similar results as in the Day 4 data in terms of which transcripts are of different abundance in each species, and in this case the number of reads mapping to each parasite transcriptome are much more similar.

The biological significance of the new finding that merozoite mRNAs are more expressed in Hh is not clear. The three highest expressed merozoite mRNAs are also significantly expressed in the Hh sporozoite (e.g. 253250, 277230, 287040), which is a product of the definitive cycle. It is not clear merozoite mRNA expression in these cultures represents a normal progression or slow mRNA turnover leading to carry over.

We appreciate this comment and have addressed this in the text by discussing the possibility of it being carryover for D4 and/or 24h infection samples, but that if this is the case the sporozoite RNAs would have to be stable from excystation through up to 15 days in culture. We feel that this is unlikely (but still state that we don’t, as of yet, understand the significance of this finding). (subsection “The spontaneous differentiation process in H. hammondi is characterized by the increased abundance of hundreds of known bradyzoite-specific genes”).

The lack of evidence of protein expression also reduces the significance of this study. Thus, these results may have more to do with the limitations of parasite in vitro behavior than natural biology.

Again as above we agree that transcriptome differences could be due to in vitro biological differences in RNA carryover/stability, which would certainly be of great interest if RNAs were maintained from sporozoites all the way through to D15 zoites. Moreover, it is important to note that the transcriptome is clearly changing during in vitroculture (from D4 to D15, Figure 3 and Figure 3—figure supplements 1 and 2), as is the biology of the parasite (revealed by cyst wall staining; Figure 2). This provides strong evidence that this is not an issue of parasite mortality as suggested. (An extensive discussion of these ideas/points can be found in in the Discussion section.)

Furthermore, we have no reason to expect that we would observed significant differences between transcript abundance and protein level during development as transcript abundance for the majority of genes in *T. gondii* is known to correlate with protein abundance in *T. gondii* (4), and therefore while some members of the “CAT STAGE SPECIFIC” gene sets may be

regulated post-transcriptionally, it is highly unlikely that all of them would be regulated in this manner. However, to further address this concern, we have compared our data with two recent studies investigating translational control and protein abundance in *T. gondii*. Our analysis revealed a positive correlation between changes in transcript abundance between D4 and D15 *H. hammondi* from our RNAseq analysis and changes in protein abundance between tachyzoites and cysts in *T. gondii* Pru (5) (Author response image 1). Based on these data, it almost certain that the bulk of the changes in transcript abundance that we identified will also correlate with changes in protein abundance. To further support this claim, we utilized available ribosomal (e.g., polysome)-SEQ data from *T. gondii* (6) to filter our genes into groups that have demonstrated some level of mRNA regulation and groups that are regulated at the level of translation. We then performed GSEA with these filtered genes and found similar enrichment for bradyzoite and tachyzoite gene sets as we describe in Figure 3, further demonstrating that enrichment for bradyzoite transcripts in *H. hammondi* is biologically relevant (Author response image 1). Taken together, the unprecedented differences in transcript abundance that we have described for replicating *H. hammondi* reveal mechanistic insight into regulation of the developmental program executed by these parasites. While we agree that proteomic analysis during this developmental program would help us create a more complete characterization of development, the restrictive pre-programmed life cycle of *H. hammondi* resulting in limited parasite growth makes it highly likely that experiments of this nature with be overwhelmingly unsuccessful. We provide these data below, and have also added a discussion of this point in the text.

**Author response image 1. respfig1:** Additional analysis of identified differences in transcript abundance between *H*. *hammondi* Eth1 and *T. gondii* VEG on a translational level. A) GSEA of transcripts detected in our D15 analysis that are known to display some level of transcriptional regulation (see (6)). B) Log_2_ fold change of transcripts that significantly differed in abundance (Log_2_ fold change >1, padj < 0.05) between D4 and D15 in *H. hammondi* Eth1 versus differentially expressed proteins (Log_2_ fold change > 1) between tachyzoite and cysts (see (5))

Notwithstanding, the obvious technical issues, the main conclusion of these experiments, is that Hh expresses bradyzoite mRNAs earlier and at higher levels compared to Tg, which was published by this group in 2014.

In the 2014 study we performed microarray analyses on RNA extracted from sporulated oocysts. In the present study we performed RNAseq on parasites that were grown in culture for 24 hour, 4 days, and 15 days. The fact that the observed higher abundance of bradyzoitespecific transcripts occurs in *H. hammondi* at D4 and then becomes even more extensive at D15 post infection is not trivial. Given the difficulties of obtaining significant amounts of *H. hammondi* RNA from in vitro cultures, RNAseq is the most viable approach to examine developmental changes that occur, and these data shown that while many “bradyzoite” genes are of higher abundance in *H. hammondi* zoites even 24 h after excystation, there are a number of others that change dramatically over the course of in vitro culture. Please see subsection “*H. hammondi* possesses unique life stages” for a discussion of this.

Given all the problems, this data needs to be consolidated and a much better discussion of the data and its caveats is also needed. Figure 8 and 9 could be combined into a single figure eliminating the outdated heat maps as they do not represent all the data and are redundant. Providing more organization and gene list breakouts along with more explanation of the columns and methods in Supplementary file is what is needed here.

We appreciate this suggestion. To improve the manuscript along these lines, we have condensed and combined the data from the previous versions of Figures 8 and 9 with our additional RNAseq data in THP-1 cells. This data is now presented in Figure 3 and associated supplementary figures. As for the heat maps, we include these representations of the data to provide multidimensional information across multiple time points, as well as the actual gene products that are being queried, in a very compact format which some of our readers will find important and interesting since many of the genes shown are known to be bradyzoite-specific. In our revision we have made sure that they are not redundant with data we have already presented. At the reviewers suggestion we have provided additional descriptive column labels and annotations in the supplementary files and added columns which indicate which significantly changed genes belong to which gene set(s) as suggested so that the reader can analyze and view the data themselves in two addition supplementary files.

References

1) Dubey JP. Oocyst shedding by cats fed isolated bradyzoites and comparison of infectivity of bradyzoites of the VEG strain *Toxoplasma gondii* to cats and mice. J Parasitol. 2001;87(1):215-9.

2) Mota CM, Chen AL, Wang K, Nadipuram S, Vashisht AA, Wohlschlegel JA, et al. New molecular tools in Neospora caninum for studying apicomplexan parasite proteins. Sci Rep. 2017;7(1):3768.

3) Ma L, Liu G, Liu J, Li M, Zhang H, Tang D, et al. Neospora caninum ROP16 play an important role in the pathogenicity by phosphorylating host cell STAT3. Vet Parasitol. 2017;243:135-47.

4) Xia D, Sanderson SJ, Jones AR, Prieto JH, Yates JR, Bromley E, et al. The proteome of *Toxoplasma gondii*: integration with the genome provides novel insights into gene expression and annotation. Genome Biol. 2008;9(7):R116.

5) Wang ZX, Zhou CX, Elsheikha HM, He S, Zhou DH, Zhu XQ. Proteomic Differences between Developmental Stages of *Toxoplasma gondii* Revealed by iTRAQ-Based Quantitative Proteomics. Front Microbiol. 2017;8:985.

6) Hassan MA, Vasquez JJ, Guo-Liang C, Meissner M, Nicolai Siegel T. Comparative ribosome profiling uncovers a dominant role for translational control in *Toxoplasma gondii*. BMC Genomics. 2017;18(1):961.

Reviewer 2 Minor Comments:

1) It looks like replication goes for a set number of divisions before terminal differentiation. This is why transfer can be achieved but the program is still counting and the expansion post transfer correlates to the original start of the initial culture and does not reset to day 1 on transfer. This should be more clearly explained in the text

We have added additional explanation of the link between number of divisions and terminal differentiation.

2) As to T. gondii. Bradyzoites found in in vitro cysts exist in both Go and G1 stages. The G1 stages can replicate in cell cultures and resinfect the intermediate host and it is likely the G0 are limited to cat infection (similar to Harmmondia). This is supported by plaque assays from mouse brain cysts, only a subset of zoites that are counted in a heamocytometer can establish infection in vitro.

This is an interesting idea and one that we will possibly pursue in future experiments. We discuss this in the revised submission.

3) For the mouse experiment is there any data on expansion of the infection (e.g. is the final parasite load dependent on how long the culture has been in vitro) One would expect with a fixed division differentiation program this would be the case.

This is an interesting suggestion and one that we will certainly consider. However we feel that this is outside the scope of the present work in that it would require initiating controlled in vivo infections and developing new assays to quantify parasite burden in a variety of tissues. We do discuss these ideas and how they could be used to validate this model in vivo.